# PROSPECT PTMs: Rich Labeled Tandem Mass Spectrometry Dataset of Modified Peptides for Machine Learning in Proteomics

Wassim Gabriel [1*]     Omar Shouman [1*]     Ayla Schroeder [1]     Florian Boessl [1]

Mathias Wilhelm [1,2]
{firstname.lastname}@tum.de

[1] Computational Mass Spectrometry
School of Life Sciences
Technical University of Munich
Freising, Germany

[2] Munich Data Science Institute (MDSI)
Technical University of Munich
Garching, Germany

## Abstract

*Post-Translational Modifications* (PTMs) are changes that occur in proteins after synthesis, influencing their structure, function, and cellular behavior. PTMs are essential in cell biology; they regulate protein function and stability, are involved in various cellular processes, and are linked to numerous diseases. A particularly interesting class of PTMs are chemical modifications such as phosphorylation introduced on amino acid side chains because they can drastically alter the physicochemical properties of the peptides once they are present. One or more PTMs can be attached to each amino acid of the peptide sequence. The most commonly applied technique to detect PTMs on proteins is bottom-up Mass Spectrometry-based proteomics (MS), where proteins are digested into peptides and subsequently analyzed using Tandem Mass Spectrometry (MS/MS). While an increasing number of machine learning models are published focusing on MS/MS-related property prediction of unmodified peptides, high-quality reference data for modified peptides is missing, impeding model development for this important class of peptides. To enable researchers to train machine learning models that can accurately predict the properties of modified peptides, we introduce four high-quality labeled datasets for applying machine and deep learning to tasks in MS-based proteomics. The four datasets comprise several subgroups of peptides with 1.2 million unique modified peptide sequences and 30 unique pairs of (amino-acid, PTM), covering both experimentally introduced and naturally occurring modifications on various amino acids. We evaluate the utility and importance of the dataset by providing benchmarking results on models trained with and without modifications and highlighting the impact of including modified sequences on downstream tasks. We demonstrate that predicting the properties of modified peptides is more challenging but has a broad impact since they are often the core of protein functionality and its regulation, and they have a potential role as biomarkers in clinical applications. Our datasets contribute to applied machine learning in proteomics by enabling the research community to experiment with methods to encode PTMs as model inputs and to benchmark against reference data for model comparison. With a proper data split for three common tasks in proteomics, we provide a robust way to evaluate model performance and assess generalization on unseen modified sequences.

---

*Equal Contribution.

38th Conference on Neural Information Processing Systems (NeurIPS 2024) Track on Datasets and Benchmarks.

# 1   Introduction

Proteins are fundamental components of living organisms, performing diverse biological functions essential for cellular processes, signaling, and structural integrity. The field of proteomics aims to study and understand the complex landscape of proteins present within a biological system. To facilitate analysis and identification, proteins are typically digested into smaller components, called peptides, using techniques such as enzymatic digestion [1]. In the widely used bottom-up proteomics approach, peptides serve as the primary unit for investigation and characterization using Mass Spectrometry-based proteomics (MS), with peptides typically composed of 5 to 50 amino acids [2].

MS-based proteomics has revolutionized the study of proteins by enabling the identification and quantification of peptides in complex biological samples, facilitating the identification and quantification of proteins [3]. Throughout MS experiments, various peptide properties are captured, providing valuable information for downstream applications, including peptide sequence identification and quantification [4]. However, an important aspect that significantly influences peptide properties is the presence of post-translational modifications (PTMs) on amino acid side chains or peptide linkages.

PTMs are frequent structural changes that occur after protein translation and directly affect its function, allowing cells to respond quickly to stimuli [5]. PTMs occur at distinct amino acid side chains, hence modifying its chemical structure [6]. More so than individual modifications, PTM crosstalk is a major factor in defining protein function, and various diseases have been connected to altered PTM patterns [7, 8, 9]. This makes the study of PTM regulation extremely valuable for both fundamental and applied clinical research [10], including drug response [11]. Finally, PTMs impact a wider scope of tasks and applications such as protein folding [12, 13] and drug reception [14].

PTMs were discovered over an extended period of time, highlighting their abundance and diversity in biological proteomes. With over 400 PTMs discovered until now [6], PTMs play an essential role in understanding proteins and their functions. PTMs are instrumental in understanding cellular processes, aging, and diseases. They occur naturally with a varying level of abundance (see Appendix Section C); some PTMs, such as phosphorylation and acetylation, are very frequent, while others, like sumoylation, are relatively rare [6]. These modifications can be naturally occurring or be induced by the researcher or the experimental setup. To facilitate experimentation, researchers introduce modifications such as Cysteine carbamidomethylation and Tandem Mass Tags (TMT), which is a technique for sample multiplexing used to concurrently analyze up to 18 separate samples in a single MS run [15, 16]. Other modifications are introduced unintentionally, such as Oxidation.

PTMs modify multiple peptide properties, including but not limited to mass, charge, retention time, and fragment ion intensities. Copies of peptide sequences can be present in different modification states. Each variant has modifications at different locations and displays different properties. Hence, it is crucial to incorporate PTM information into machine learning tasks to accurately predict properties of peptide sequences. The choice of how to represent sequences and the present modifications on the amino acids dictates the usage of specific model architectures. Although sequence-based models are relatively dominant [17], Graph Neural Networks can be applied to some problems since amino acids and modifications can be represented as molecular graphs [18, 19, 20].

Datasets and machine learning tasks to predict properties of peptide sequences are well established in the proteomics research community [21, 17]. However, incorporating PTM information in prediction tasks is still in its early phase, specifically when it comes to labeled reference datasets comprising sequences rich in PTMs. Such datasets would facilitate training and benchmarking models with different ways of incorporating PTMs into a machine learning workflow.

Protein Language Models (pLM), leveraging the embeddings from pre-trained language models on large protein databases, are common in down-stream tasks [22, 23], such as PTM site prediction [24]. However, pre-trained pLMs do not typically capture PTM information and focus on protein sequences rather than peptides, hindering the development of down-stream PTM-related tasks.

Here, we introduce four annotated datasets, rich with modified peptide sequences, including 1.2 million unique modified peptide sequences and 30 unique amino acid-PTM combinations (i.e. distinct pairs of amino acid and PTM); more details are shown in Appendix Section A.2 in Table 3. These amino acid-PTM combinations are found in the human proteome and can be mapped back to human proteins. The datasets are based on upstream raw data available in ProteomeTools, a synthetic human proteome measurement [25, 26, 27] as well as novel unpublished data. The novelty of the datasets

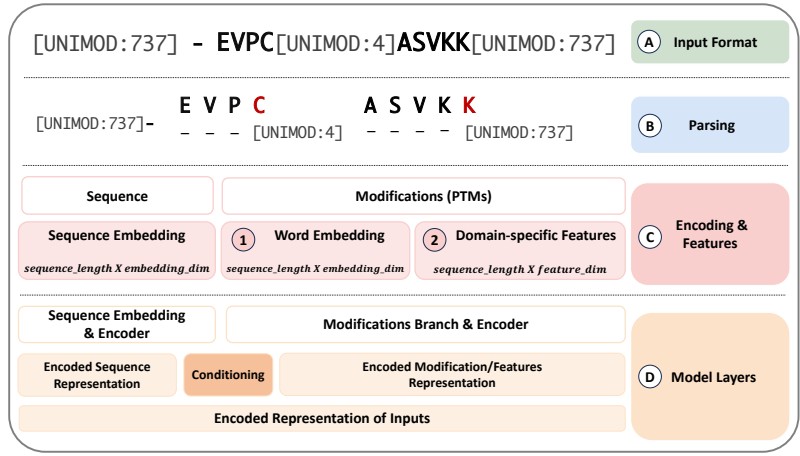

Figure 1: Sequence life cycle with modifications as inputs to a deep learning model.

is two-fold: the new mass spectrometry data with PTMs and the rich annotations provided. The annotated and processed datasets are aimed to complement and extend the original PROSPECT dataset, which primarily covers various classes of unmodified peptides [21, 28]. The datasets target primarily three tasks in proteomics; retention time (RT), fragment ion intensity, and precursor charge prediction. Each newly introduced dataset includes examples representing a common subgroup of modified peptide sequences; (1) TMT dataset includes peptide sequences labeled with Tandem Mass Tags, (2) Multi-PTM includes examples rich with 13 unique (amino-acid, PTM) pairs, (3) TMT-PTM includes six unique pairs of (amino-acid, PTM) occurring in TMT-labeled sequences. Our final dataset, Test-PTM, comprises 21 unique pairs of (amino acid, PTM) and is well-suited as a hold-out/test dataset. For example, it includes modifications not present in the other three datasets. This characteristic, along with others we detail in section 3.3, makes Test-PTM a good candidate for evaluating and benchmarking models that can encode, process, and learn useful representations of PTMs to predict various peptide properties.

## 2 Related Work

Inspired by models from Natural Language Processing (NLP) that can process sequential data, proteomics researchers developed and trained various model architectures, following a similar workflow as highlighted in [21]. We summarize previous work in literature that tackled PTMs and highlight the different ways of encoding and feeding PTMs as inputs to deep learning models. Figure 1 depicts the life cycle of the input data (peptide sequences with PTMs present) with a focus on representing PTMs. The explained workflow can be considered a common ground for several supervised learning tasks, specifically for the three tasks: retention time, fragment ion intensity, and precursor charge state prediction.

### 2.1 Input Format and Parsing

Input sequences are stored in a structured file format as strings. Each sequence is composed of amino acids with PTMs present at specific locations (sites) of the sequence. Amino acids are typically represented with a widely-adopted one letter code based on the IUPAC nomenclature [29], while modifications are represented by either a short name or an ID coming from common ontologies or controlled vocabularies [30, 31, 32]. An example sequence is shown in the input parsing step in Figure 1 part A, where modifications are represented by their Unimod ID [30].

Presenting input sequences in one field together with modifications at their respective locations is used in some models in the literature, such as Prosit [33] and PrositTransformer [34]. Opting for this approach reduces the assumptions on input data but implies an additional parsing step in the data preparation before training the model. For parsing sequences and modifications presented as one field, researchers either leverage existing Python packages such as Pyteomics [35, 36] or write their own utility and helper functions. Figure 1 part B illustrates what the parsing step would produce.

For easier processing of modified sequences, several published models require the input data to be provided in two separate fields: one with the sequence of amino acids and the other with the modification name and site (i.e., index of the amino acid on which the modification is present). This approach is followed by DeepLC [37], pdeep2 [38], pdeep3 [39], and AlphaPeptDeep [40]. Parsing and loading data is simpler in this case since indices with modification sites and names are readily available in a dedicated field in the input. Yet, for inference, raw data has to be prepared accordingly.

## 2.2 Encoding and Representation of Sequences with Domain-Specific Features

A short review of encoding methods for input sequences with PTMs reveals two main options for representing modifications as additional inputs aligned with the peptide sequence. The first option involves encoding each modification using one-hot encoding or word embedding. Using embeddings allows for learning representations for all modifications present in the training data, as implemented in [41]. The second option is to parse the modifications before training and represent them with extracted features based on domain knowledge and the changes they introduce to the amino acids. These features can include changes in chemical composition, mass, or other relevant properties of the modified amino acids. This approach is present in DeepLC [37], pDeep3 [39], and AlphaPeptDeep [40]. Both alternatives are illustrated in Figure 1 part C.

Using word embeddings for PTMs provides a way to learn numeric representations for known modifications, which is well-suited for the supervised setup. Moreover, with a sufficiently large amount of training data, the subsequent layers in the model would learn to combine inputs to provide robust features relevant to the task, eliminating the need for manually designed feature extractors [42]. However, since embedding vectors are learned from the training dataset and the modifications it contains, models would suffer from the out-of-vocabulary problem. Therefore, prediction of peptide sequences incorporating unseen modifications becomes challenging or is of relatively low quality. In contrast, feature extraction methods based on domain knowledge solve this problem by extracting relevant domain-specific information about the modifications from databases like Unimod [30] using tools such as Pyteomics [35, 36]. This approach ensures that unseen modifications can be represented and fed to the model by extracting the relevant features at inference. Nevertheless, to hand-design good features, a considerable amount of domain expertise and time are required compared to automatically learning features [42]. For example, DeepLC [37] incorporates atom counts at each amino acid after modifications and the sum of atom counts at neighboring pairs of amino acids as extracted features to improve retention time prediction of PTM sequences.

Several model implementations in the literature limit the scope of their training and inference to specific modifications to reduce the complexity of handling PTMs. For instance, Prosit-TMT [43], pDeep3 [39], and DeepFLR [41] have focused on particular types of PTMs. These models leverage the characteristics of the target modifications to improve prediction performance.

## 2.3 Model Architectures to Process Modifications

To process PTMs effectively, deep learning models typically include a dedicated branch in the architecture to encode the modifications (a sequence aligned with the amino acid sequence without modifications) or process the extracted features. For instance, DeepLC [37] includes a distinct branch of convolutional layers to consume the extracted features from the modifications. Models that encode modifications with word embeddings, as the case for DeepFLR [41], usually have a trainable embedding layer for PTMs. For both approaches, models combine the encoded representations of the amino acid sequence and the sequence of modifications using a conditioning technique. The most common conditioning methods applied are element-wise multiplication as in Prosit [33], concatenation as in DeepLC [37], and matrix addition as in DeepFLR [41]. Encoding and conditioning are depicted in Figure 1 part D. Depending on the model architecture, conditioning can happen early in the model (directly after the embedding layer) or later after encoding each input in a separate branch (sequence and modifications).

## 2.4 Challenges

In the traditional database search, peptides and their modifications are identified by comparing their experimental mass spectra (MS2) with theoretical spectra generated from a database of protein sequences [44]. However, as the number of potential modifications increases (due to multiple PTM

Table 1: Summary statistics of the dataset.

| Dataset | Packages | Pools | Unique Peptides | Precursors | Spectra | Annotated Peaks | Raw Peaks |
|---------|----------|-------|-----------------|------------|---------|-----------------|-----------|
| TMT | 11 | 1000 | 714 K | 820 K | 28.2 M | 1.8 B | 11.2 B |
| Multi-PTM | 12 | 388 | 306 K | 412 K | 19.6 M | 2 B | 6 B |
| TMT-PTM | 6 | 260 | 118 K | 132 K | 7 M | 456 M | 2.8 B |
| Test-PTM | 29 | 147 | 53 K | 71.5 K | 3.9 M | 248 M | 936 M |

types and potential sites), the search space for possible peptide modifications expands dramatically. This search space explosion can result in more potential matches due to the higher chance of collisions, where multiple peptides could have similar mass/charge. Hence, it becomes more challenging to confidently match precursor ions to peptide sequences. Consequently, search engines might consider more possible peptide matches for the MS2 spectra, leading to difficulties separating correct from incorrect matches. Moreover, precisely determining the location of a PTM within a peptide sequence poses a challenge primarily attributed to the possible occurrence of the PTM at multiple potential residues, as the MS2 spectra for these different permutations are similar [41]. In the meantime, accurate localization of PTMs is crucial, as biological functions are often associated with different PTM sites [45]. While several tools try to improve the accuracy of PTM localization, ground truth datasets like the ones we are introducing here would help improve such tools [41, 46, 47].

## 3 Dataset

Similar to the version of PROSPECT with unmodified peptide sequences [21, 28], ProteomeTools [48, 49] is the base for our four datasets. It contains multimodal liquid chromatography-tandem mass spectrometry analysis for over a million synthetic peptides, representing all canonical human gene products. We leverage the raw data from ProteomeTools and provide annotations and labels for retention time, charge, and MS2 spectra. Our choice of ProteomeTools as an upstream raw dataset is motivated by three reasons: (1) several models in literature use it for training models [33, 34, 37, 38, 50, 51] and for finding diagnostic features for PTMs [52], (2) it contains high-quality spectra with PTMs already localized, (3) all measurements are from synthetic samples of peptides. We refer to ProteomeTools [48, 49] and PROSPECT [21] for more details on the advantages of using ProteomeTools as an upstream dataset.

Our datasets are designed to facilitate training and evaluating machine learning models on various tasks in MS-based proteomics, particularly those involving modified peptide sequences. Since modifications change the chemical structure of the target amino acid, they impact the predicted properties of the peptide sequences. Due to the altered properties of the peptide sequences, the presence of modifications adds complexity to prediction tasks, making them more challenging. Furthermore, to support PTM localization tasks [53], we included peptide sets with permuted phosphosites (more details in Appendix Section A).

This section briefly describes the schema of the datasets, provides summary statistics and exploratory analyses focusing on PTMs, and highlights the impact of the introduced datasets. We primarily focus on retention time, fragment ion intensity, and precursor charge state prediction, yet the datasets can be used for several tasks in proteomics. Appendix Section E presents a full list of supported tasks.

### 3.1 Dataset Generation and Schema

We apply the same annotation workflow as in the original PROSPECT dataset [21], which is based on an expert annotation system [54] to annotate $y$ and $b$ fragment ions (up to triple-charged) as well as possible neutral losses. While retention time and precursor charge states are directly extracted from the MS experiments. Our implementation of the annotation pipeline is available in a dedicated GitHub repository under the name Spectrum Fundamentals [55].

The four datasets comprise 61 packages, each with two main parquet file formats: meta-data and annotation files. Each package has one meta-data file, while the annotations file is split into multiple files per package to facilitate reading the data. Annotation files are sub-organized by pools, where a

pool is a set of ∼ 1k peptides measured in one analysis run. A unique identifier is provided in both files to trace any example to its original raw data file in ProteomeTools. This identifier combines the raw file ID and the scan number. The original ProteomeTools upstream datasets are partially available on PRIDE [25, 26, 27] and have the same identifier names. Our annotated datasets are fully available online and are hosted on Zenodo [56, 57, 58, 59]. Figure 2 shows the various datasets we make available with this publication. Moreover, it explains the applied steps to the raw upstream datasets to arrive at the full Zenodo annotated datasets and eventually to the processed and split HuggingFace dataset for each of the three tasks in focus. Table 1 shows summary statistics of the data. The modified peptide sequences are represented as strings, allowing flexibility in using the dataset in different encoding and machine learning pipelines. To represent the sequences, we follow one of the recommended notations of ProForma [60], a standard notation for writing sequences with modifications. More statistics about the data are in Appendix Section C.

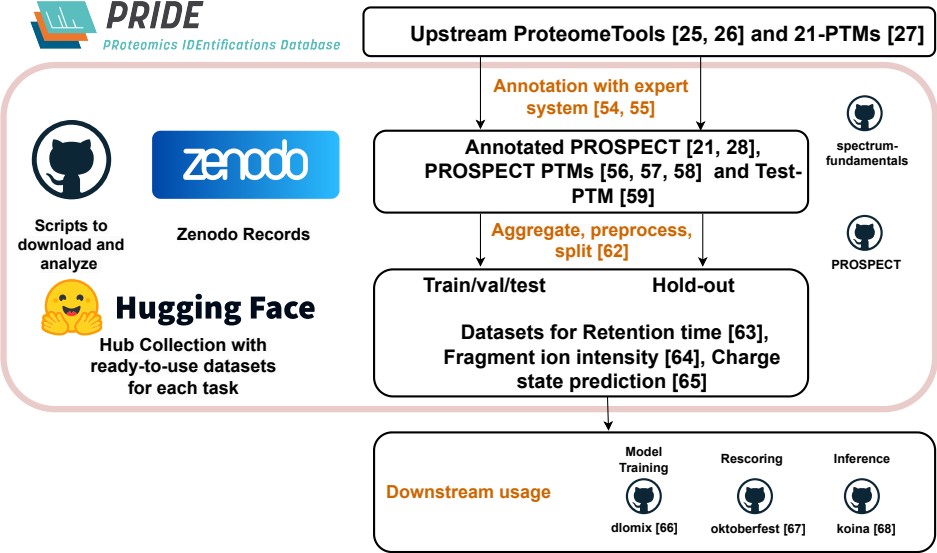

Figure 2: Datasets made available with our publication, with upstream raw data sources and downstream usage highlighted.

## 3.2 Exploratory Data Analysis

To understand the diversity and distribution of modifications in our datasets, Figure 5 in Appendix Section C shows a heat map displaying the frequency of PTMs (log scale) on the respective amino acid sites. Figure 6 illustrates horizontal bar plots with log scale counts of the occurrence of PTMs and modified amino acids. Several observations indicate the complexity of handling PTMs, including (1) some PTMs occur more frequently than others, (2) PTMs occur only on specific amino acids, and (3) some amino acids are more frequently reported as modification sites in comparison to others.

## 3.3 Data Split

As highlighted in Table 1, we curated four datasets, each including a subgroup of modified peptide sequences. We recommend using the first three datasets for training and validation splits during model training and hyper-parameter optimization, where data splitting should follow a sequence-based disjoint split. The Test-PTM dataset can solely be used to evaluate the model and its performance on unseen examples. This choice is motivated by several reasons: Test-PTM contains (1) sequences with the same PTM occurring at the same residue as in the other datasets but at different sites within the sequence, (2) sequences with the same PTMs as in the other datasets occurring at different residues (3) sequences with multiple PTMs occurring at different residue sites in the same sequence, (4) modifications that are not present in the other three datasets, (5) sequences with permuted phosphosites with their experimentally-acquired fragmentation spectra A.1, and (6) unmodified counterparts of modified peptide sequences. These characteristics allow for quantifying model performance on

sequences with PTMs and performance comparison between modified and unmodified peptide sequences. Further details on splitting the data are in Appendix Section G.

For accessibility to the machine learning community, we provide the preprocessed and filtered datasets with the recommended split for each of the three tasks on the Hugging Face Hub [63, 64, 65].

## 3.4 Dataset Utility

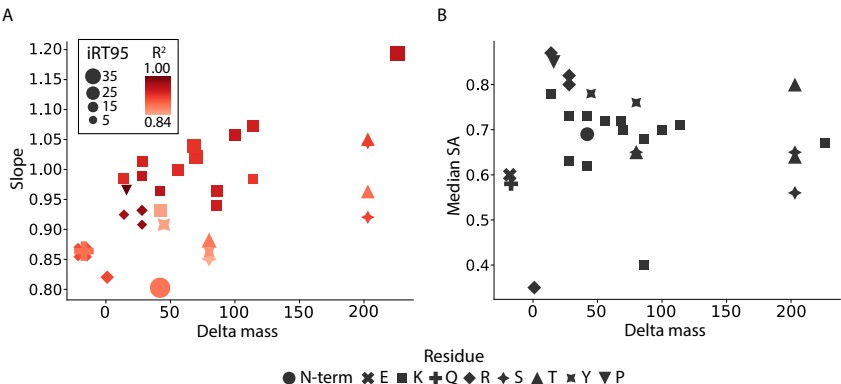

Figure 3: Scatter plots summarizing differences in iRT and fragment ion intensity between modified and matching unmodified peptides.

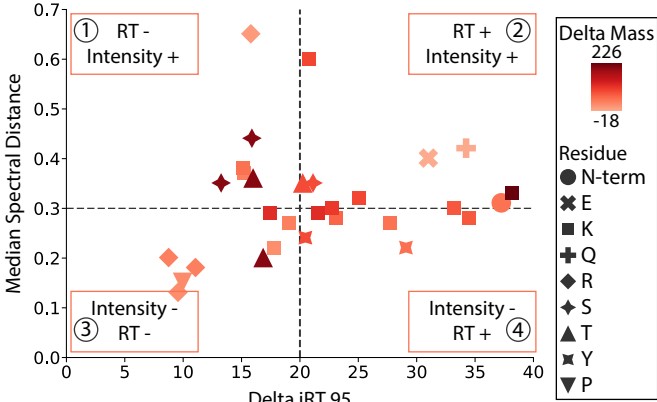

Figure 4: Impact of different PTMs on retention time and the fragment intensities compared to its unmodified peptide counterparts. A spectral distance of 0.35 and delta iRT of 20 are commonly observed cutoffs in rescoring when differentiating correct from incorrect matches.

The utility of the curated datasets can be highlighted by looking at the impact of PTMs on peptide properties, such as retention time and intensity of the fragment ions in the spectrum. We use common evaluation metrics specific to each property to compare modified and unmodified sequences, such as time-delta at $95\%$, $R^2$ and slope for retention time prediction, and the Spectral Angle (SA) for fragment ion intensity prediction [21]. SA only captures the difference in the peak intensity and not the mass shift introduced by different PTMs. We compute the metrics with the experimental data (ground truth labels) for each task, the indexed Retention Time (iRT), the vector for fragment ion intensity, and the observed precursor charge state distribution. More details are in Appendix D.

Figure 3 shows the relationship between the change in mass induced by the different PTMs versus the effect on peptide properties quantified by the respective metric. For RT, panel A shows the slope versus the introduced change in mass. Panel B depicts the SA versus the change in mass to underscore the impact of PTMs on the MS2 spectra. Except for a limited number of modifications on lysine (K), other modifications do not exhibit a linear correlation between mass and the respective change in peptide properties. The missing correlation indicates that a simple feature as m/z would not help

improve predictions; therefore, better approaches to encode PTMs and encapsulate the complexity of their impact are essential to improve predictions for peptide properties. Figure 4 shows the change in iRT and spectra induced by different PTMs to highlight that modifications can change properties in different ways. In quadrants two and three, similar patterns are observed for both properties, wherein the impact is either significant or insignificant. Conversely, quadrants one and four exhibit dissimilar behavior, with one property experiencing a substantial change while the other experiences a minimal one.

Peptides often occur in more than one precursor charge state, with the possible charge states of a peptide being called charge state distribution (CSD) [69, 70]. Next to affecting a peptide's RT and fragment ion intensity, PTMs are also known to alter a peptide's CSD [70]. Training accurate CSD prediction models can enable a meaningful reduction of the size of predicted proteome-wide spectral libraries, thus increasing their specificity. One shortcoming of the PROSPECT PTM datasets is that they mostly contain peptides in a single precursor charge state, as depicted in Figure 7 in Appendix section C.

To further demonstrate the impact of the introduced datasets, we present an explicit example of a downstream task, showing the improvement of PSMs (Peptide-Spectrum Matches) and peptide identification rates on a TMTpro (18plex) phosphoproteome dataset [16]. We chose this dataset because PROSPECT PTM datasets do not contain TMTpro-labeled peptides, and we wanted to showcase the multi-PTM aspect of trained models. More details are in Appendix Section F. Besides Prosit [33], many other peptide property prediction models, which also generalize well on downstream tasks, were trained or evaluated on the ProteomeTools data, like pDeep [38], AlphaPeptDeep [40], SpecEncoder [71] or InstaNovo [72], underlining the importance of the datasets and the proposed extension to PTMs. Because of the lack of PTM datasets, most models only explicitly support a small number of PTMs, and thus, our benchmarking highlights the research gap. In the example of fragment ion intensity prediction, while some models claim to support more PTMs, this is often only achieved by shifting the respective fragment ions in m/z space and not addressing the impact of the PTM on intensities. For precursor charge state prediction, currently, no published model can predict more than a few predefined PTMs to the best of our knowledge. Furthermore, PROSPECT PTM datasets are not limited to applications for boosting identifications. Peptide property predictions find applications in spectral library generation that would greatly benefit from additional predictors to reduce the library size, such as precursor charge state. This is critical, for example, in metaproteomics experiments where analysis suffers from huge search space [73]. A smaller library size is preferred to increase sensitivity and specificity while reducing the computational cost. Peptide property predictions can also be used for single peptides, e.g., in targeted proteomics experiments, to pick the best collision energies [74], or to validate single peptide identifications by visually comparing experimental vs. predicted fragmentation spectra [75], e.g., in immunopeptidomics for neo-antigen validation, all relevant for translating findings to medicine [76].

## 4 Evaluation

Improving peptide identification is one of the main objectives of accurately predicting peptide properties. As we illustrated, predicting properties of peptides with PTMs is more challenging and requires incorporating PTM information. To evaluate models trained on modified sequences for predicting retention time and MS/MS spectra, we choose two intrinsic evaluation metrics that quantify the model performance. As a baseline, we report prediction results from a Prosit model [33] trained on unmodified sequences and compare them against the experimental values (true labels) to highlight the importance of incorporating PTM information. At inference time, PTM information is not taken into account since the model was trained on unmodified sequences. We additionally report prediction results for retention time on DeepLC [37] and AlphaPeptDeep [40] for MS2 spectra since they incorporate PTM information based on atom counts from the chemical structure of PTMs and amino acids.

Table 2 summarizes the performance of the different models used for benchmarking the three tasks with the recommended evaluation metric, with metrics reported on Test-PTM dataset, split in seen/unseen PTMs during training (relevant only to Prosit [33]). The Prosit [33] naive model was trained on all datasets except Test-PTM. Hence, it does not support unseen PTMs but rather ignores them. Two variants of Prosit encoded PTMs with domain-specific features: Prosit-DeltaMass and Prosit-DeltaAtoms. Prosit-DeltaMass uses the mass introduced by the PTM as an input feature to the

Table 2: Model performance comparison for the three tasks.

| Model PTMS | Retention time (Timedelta 95) | | Charge state (MAE) | | Fragment ion intensity (Spectral Angle) | |
|---|---|---|---|---|---|---|
| | unseen | seen | unseen | seen | unseen | seen |
| Prosit baseline [33] | 26.1 | 25.8 | 0.17 | 0.16 | 0.75 | 0.74 |
| DeepLC [37] | 19.5 | 13.7 | – | – | – | – |
| AlphapeptDeep [40] | 14.2 | 12.7 | – | – | 0.78 | 0.83 |
| Prosit naive-encoding | 25.2 | 16.4 | – | – | 0.79 | 0.89 |
| Prosit - DeltaMass [77] | 17.2 | 14.2 | – | – | 0.81 | 0.87 |
| Prosit - DeltaAtoms [77] | 12.7 | 9.9 | – | – | 0.86 | 0.89 |

model, while Prosit-DeltaAtoms uses the atom count introduced by the PTM [40, 37]. AlphaPeptDeep [40] does not support N-term modifications.

The metrics show that the base model performs poorly on modified sequences since it does not encode present modifications. Improved performance is observed for Prosit naive-encoding only on seen modifications. Prosit-DeltaMass performs better than the base model but does not fully capture the nuances of PTMs since several PTMs with similar mass behave differently. Prosit-DeltaAtoms performs best across all model variations as it encodes PTMs with more complex domain-specific information. More experimental details can be found in Appendix Section H.

### 4.1 Retention Time Prediction

We use the time delta iRT metric for RT, which provides fine-grained and domain-specific insights into the model performance [51, 33, 37]. As recommended by PROSPECT [21], we report the time-delta at $95\%$ $\Delta t_{95\%}$, which is the minimal time window containing the errors (residuals) between observed and predicted retention times for $95\%$ of the peptides [78]. An implementation of this metric is available at the GitHub repository [62].

Figure 10 in Appendix Section D.2 shows in more detail the distribution of retention time predictions across different PTMs, comparing our best performing model (Prosit-DeltaAtoms) to DeepLC [37], one of the state of the art retention time PTM-aware models.

### 4.2 Fragment Ion Intensity Prediction

For MS/MS spectra, we use the normalized spectral angle used in Prosit [33] and recommended in PROSPECT [21]. Code for calculating the spectral angle is available at the GitHub repository [62].

Figure 11 in Appendix Section D.2 shows in more detail the distribution of fragment ion intensity predictions across different PTMs, comparing our best performing model (Prosit-DeltaAtoms) to AlphaPeptDeep [40], one of the state of the art fragment ion intensity PTM-aware models.

### 4.3 Precursor Charge State Prediction

We use the mean absolute error (MAE) for precursor charge state prediction. In contrast to the Pearson Correlation Coefficient (PCC) used in previous precursor charge state prediction publications [69, 70], MAE can robustly assess cases where peptides occur only in a single or two distinct charge states. Figure 13 in Appendix Section D.2 shows the distribution of precursor charge state predictions across different PTMs in more detail.

## 5 Conclusion and Limitations

This work introduced PROSPECT PTMs, four annotated datasets for MS proteomics research based on ProteomeTools [48, 49]. The datasets contain peptide sequences with various PTM types occurring at different amino acid sites. Although the datasets are not limited to retention time, fragment ion intensity, and precursor charge state prediction, we focused on these three tasks due to their importance in downstream applications [79, 80, 67]. We recommended metrics and visualizations for model

evaluation, especially when incorporating PTM information. The Hugging Face Hub versions of the datasets are processed and split to provide all the required annotations for the three tasks to train and evaluate machine learning models [63, 64, 65]. We provide benchmarking results for six in-house novel trained and recent state-of-the-art deep learning models. The models include Prosit [33] as a baseline pre-trained on unmodified peptides and three variants of Prosit trained on modified peptides: Prosit-Naive, Prosit-DeltaMass, and Prosit-DeltaAtoms. Additionally, we report results on DeepLC [37] for retention time prediction with PTMs and AlphaPeptDeep [40] for both retention time and fragment ion intensity prediction with PTMs.

Although the datasets include examples for various PTMs as highlighted in Appendix Section C, we acknowledge the limitations implied by omitting others. However, these limitations are inherent to the biological origin of the data; new PTMs are still being discovered [81, 82], and only some PTMs can be synthesized efficiently on a large scale. Using experimental data for others bears the risk of generating a training set with an unknown number of false positives. Another limitation is that the mass analyzers used to acquire the datasets only cover Orbitraps and Iontraps. Nevertheless, our experiments showed that models trained on those (e.g., Prosit [33]) generalize to TimsTOF data and lead to similar increases in peptide identifications [83]. We expect further data examples from other peptide sets (e.g., PTMs) and mass spectrometers (e.g., Waters) to be added over the next years, reducing biases and covering additional experimental settings.

Our annotation pipeline has a few limitations; first, we kept all annotations found for the same peak, which partially led to over-annotation. Second, the same label can be assigned to multiple peaks if they lie within the tolerance of the theoretical mass-to-charge ratio (m/z). Despite these limitations, the datasets still contain all the information required to develop better filtering approaches. A final limitation is that we restricted our annotation pipeline to a subset of possible annotation ions, leaving out some known ones, mainly diagnostic ones and ammonium ions.

The evaluation of our models on three tasks demonstrates their utility and effectiveness in extending existing models to tackle PTMs when predicting peptide properties. The results indicate that incorporating PTM information is required to improve the accuracy of model predictions, especially since model performance may vary depending on the presence of PTMs and their impact on the predicted properties. Future research should explore the development of specialized models tailored to encode and process PTMs.

One future direction is to explore and utilize augmentation techniques to enhance the generalization of models on rare or unseen modifications. Data augmentation can be useful in cases where certain modifications or amino acid sites are rarely present in the data but can be artificially introduced by augmentation. Future work should focus on establishing such techniques, utilizing our reference datasets to increase the robustness and versatility of models in handling a wider range of PTMs.

PTMs do not only alter the chemical properties of peptides but also significantly impact other characteristics, such as precursor charge and the behavior of fragment ions during analysis. Some of the most notable PTMs, such as phosphorylation, citrullination, and malonylation, result in the loss of specific chemical groups from the peptide structure [49]. These losses can lead to changes in the intensities observed during analysis, as a significant portion of the intensity is transferred to the same peak undergoing neutral loss. This phenomenon can profoundly influence the interpretation of spectra and identification of peptide sequences. The newly introduced datasets complement and synergize with PROSPECT [21], enabling the study of the change in additional less-investigated properties, such as precursor charge, thereby enhancing the depth and scope of PTM analysis. Additionally, the datasets can be used in tandem in novel ways, such as learning spectrum embeddings for matching pairs of modified and unmodified peptides, learning joint embeddings for both spectra, or training models that convert one representation to the other [84, 85, 86, 87].

Finally, our datasets open space for developing new models that encode and process PTMs, a novel area of applied research in proteomics. Additionally, it can be used to learn spectra from peptide SMILES [88] representation or vice versa, similar to the work done for small molecules in [89]. Further benchmarking and comparison with new and emerging models should be pursued to advance the field and drive improvements in the prediction performance of peptide properties with PTMs. Our reference datasets can serve as a foundation for benchmarking and comparison of new models.

## Acknowledgments and Disclosure of Funding

This work was in part funded by the German Federal Ministry of Education and Research (BMBF; Grant No 031L0008A), European Union's Horizon 2020 Program under Grant Agreement 823839 (H2020-INFRAIA-2018-1; EPIC-XS) and ERC Starting Grant 101077037.

We are grateful to Zenodo [90] for their support and cooperation in hosting our datasets.

## Conflict of Interest

Mathias Wilhelm is a founder and shareholder of MSAID GmbH and OmicScouts GmbH, with no operational role in either of the two companies.

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

# Appendix

## A   Proteomics Terminology and Acronyms

### A.1   Proteomics Terminology

- Retention Time (RT): The time taken by a peptide to pass through a column. This is dependent on different peptide features such as hydrophobicity and using a column helps to separate peptides before being analysed by a mass spectrometer.

- Fragment ions: An ion formed by fragmentation of a peptide in the mass spectrometer

- $b$ and $y$ ions: The B and Y ions for a given peptide represent the two halves formed by splitting the original peptide between various amino acids.

- Neutral Loss (NL): The loss of small molecules from peptide [92].

- MS/MS: Tandem Mass Spectrometry.

- Andromeda Score: A probabilistic score assigned to a spectrum by MaxQuant [93] to indicate the certainty of the identification. A higher score indicates higher confidence.

- Amino acid side chain: The organic R group that is unique to each amino acid.

- Peptide linkage: The chemical bond between two peptides.

- Indexed Retention Time (iRT): iRT is calculated by choosing two or more reference peptides and regressing a line between their retention times and is a unit-less quantity [94].

- Permuted phosphosites: Peptides with known phosphosites where we permute the site of phosporylation (at the same residue, but different position in the peptide). An example would be P[UNIMOD:21]EPTIDE -> PEP[UNIMOD:21]TIDE,where we have acquired spectra for both variants .This is not merely permutation on the sequence only, but rather a new input-target (i.e. sequence-spectrum) pair. Thus, these permuted phosphosites are helpful to evaluate tools for modification localization, specially since phosporylation is one of the most common PTMs and fairly complex to localize properly.

- Precursors: Peptide/Charge combinations since one peptide can occur in multiple charges.

- Tryptic peptides: Proteins digested by trypsin enzyme (most commonly used) are tryptic peptides. These peptides show a distinct pattern as they always end with K or R amino acids.

- Non-Tryptic peptides: They are peptides generated by enzymes other than trypsin. These can also be peptides which are picked with different methods other than digestion such as HLA peptides [95].

- Peptide-Spectrum Match (PSM): A peptide-spectrum match (PSM) is the assignment of a specific peptide sequence to a tandem mass spectrum (MS/MS), which contains information about fragmented peptide sequences. There are various protocols for analyzing MS/MS, but the primary goal is to assign a single peptide sequence to each MS/MS spectrum in a dataset.

### A.2   Post-Translational Modifications (PTMs)

Table 3 lists the various modifications used in the datasets together with the respective Unimod ID [30], name, abbreviation, residue, change in atomic composition and change in Mass

## B   Data Annotation

As discussed in the paper, we used an expert system to annotate the MS/MS spectra, which relies on domain-specific conditional rules. Table 4 lists the rules with their original name and number as described in the original expert system publication [54].

We applied the rules by following a sequential workflow: (1) annotate one spectrum at a time, (2) generate all possible fragment ions, (3) check for matches within the tolerance specified by the expert system. For neutral losses we annotate up to 2 consecutive neutral losses.

Table 3: PTMs abbreviations and Unimod IDs.

| UnimodID | Residue | Abbreviation | PTM Name | Delta Atom Count | Monoistopic Mass (da) |
|---|---|---|---|---|---|
| 1 | N-term/K | (ac) | Acetylation | $H_2C_2O$ | 42.01 |
| 3 | K | (bi) | Biotinylation | $H_{14}C_{10}N_2O_2S$ | 226.08 |
| 4 | C | (cam) | Carbamidomethylation | $H_3C_2NO$ | 57.02 |
| 7 | R | (cit) | Citrullination | $H_{-1}N_{-1}O$ | 0.98 |
| 21 | S/T/Y | (ph) | Phosporylation | $HO_3P$ | 79.97 |
| 27 | E | (py) | Pyro-glu from E | $H_{-2}O_{-1}$ | -18.01 |
| 28 | Q | (py) | Pyro-glu from Q | $H_{-3}N_{-1}$ | -17.02 |
| 34 | K/R | (me) | Methylation | $H_2C$ | 14.02 |
| 35 | M | (ox) | Oxidation | $O$ | 15.99 |
| 35 | P | (hy) | Hydroxylation | $O$ | 15.99 |
| 36 | K/R | (dime) | Di-Methylation | $H_4C_2$ | 28.03 |
| 37 | K | (tme) | Tri-Methylation | $H_6C_3$ | 42.05 |
| 43 | S/T | (glc) | GlcNAC | $H_{15}C_8NO_6$ | 203.8 |
| 43 | S/T | (gal) | GalNAC | $H_{15}C_8NO_6$ | 203.8 |
| 58 | K | (pr) | Propionylation | $H_4C_3O$ | 56.03 |
| 64 | K | (su) | Succinylation | $H_4C_4O_3$ | 100.02 |
| 121 | K | (ubi) | Ubiquitinylation | $H_64C_4N_2O_2$ | 114.04 |
| 122 | K | (fo) | Formylation | $CO$ | 27.99 |
| 354 | Y | (ni) | Nitro | $H_{-1}NO_2$ | 44.99 |
| 737 | N-term/K | (tm) | Tandem Mass Tag | $H_{20}C_{12}N_2O_2$ | 229.17 |
| 747 | K | (ma) | Malonylation | $H_2C_3O_3$ | 86.00 |
| 1289 | K | (bu) | Butyrylation | $H_6C_4O$ | 70.04 |
| 1363 | K | (cr) | Crotonylation | $H_4C_4O$ | 68.03 |
| 1848 | K | (gl) | Glutarylation | $H_6C_5O_3$ | 114.03 |
| 1849 | K | (hy) | Hydroxyisobutyrylation | $H_6C_4O_2$ | 86.04 |

Table 4: Rules from the used expert system [54] for annotation of MS/MS spectra.

| Rule | Name | Rule | Name |
|---|---|---|---|
| 35 | b-ion series | 80 | Neutral loss at M |
| 36 | y-ion series | 81 | Neutral loss at M(Ox) |
| 44 | Charge1+ | 82 | Neutral loss at N |
| 45 | Charge2+ | 83 | Neutral loss at Q |
| 46 | Charge3+ | 84 | Neutral loss at R |
| 49 | Neutral loss at S(ph) | 85 | Neutral loss at S |
| 50 | Neutral loss at T(ph) | 86 | Neutral loss at T |
| 74 | Neutral loss at C | 87 | Neutral loss at V |
| 75 | Neutral loss at D | 88 | Neutral loss at W |
| 76 | Neutral loss at E | 97 | Priority B Rule |
| 77 | Neutral loss at I | 98 | Priority Y Rule |
| 78 | Neutral loss at K | 105 | Priority Neutral Loss Rule |
| 79 | Neutral loss at L | | |

We use multiple threads to annotate multiple raw files in parallel. We used an AMD EPYC 7452 processor with 50 cores. The total time for annotating the complete raw data of 3 TB is around 40 hours.

Our implementation of the annotation pipeline is available in a dedicated GitHub repository under the name Spectrum Fundamentals [55]. Utilities for reading and parsing the raw data are collected in a dedicated GitHub repository under the name Spectrum IO [91].

# C   Further Exploratory Data Analysis

Tables 5, 6, 5, 8 provide further statistics on the modifications that exist in each of the three datasets with PTMs. The reported counts represent the number of unique sequences in total and those with at least one modification. All peptide sequences in the TMT dataset have a TMT modification on the N-term and all occurences of lysine (K). This specific modification is excluded from the table.

Table 5: Summary statistics of TMT dataset.

| Residue | Modification | Unique Peptides | Precursors | Spectra |
|---|---|---|---|---|
| N-Term | TMT | 641 K | 742 K | 24 M |
| K | TMT | 359 K | 414 K | 12.6 M |

Table 6: Summary statistics of Multi-PTM dataset.

| Residue | Modification | Unique Peptides | Precursors | Spectra |
|---|---|---|---|---|
| K | Acetylation | 52.1 K | 71.2 K | 3.1 M |
| N-Term | Acetylation | 6.2 K | 7.3 K | 323 K |
| R | Citrullination | 3.2 K | 5 K | 558 K |
| K | Methylation | 1.9 K | 2.8 K | 160 K |
| R | Methylation | 14.8 K | 21.2 K | 1.3 M |
| S | OGalNAc | 1.2 K | 1.8 K | 185 K |
| T | OGalNAc | 2 K | 3.2 K | 301 K |
| S | OGlcNAc | 465 | 770 | 191 K |
| T | OGlcNAc | 331 | 554 | 151 K |
| S | Phosphorylation | 33.1 K | 37.8 K | 1.5 M |
| T | Phosphorylation | 14 K | 16 K | 454 K |
| Y | Phosphorylation | 31 K | 38 K | 3.1 M |
| E | Pyro-glu | 2.8 K | 3.3 K | 173 K |
| Q | Pyro-glu | 6.9 K | 8.1 K | 358 K |
| K | Ubiquitinylation | 76.2 K | 125.6 K | 3.1 M |

Table 7: Summary statistics of TMT-PTM dataset.

| Residue | Modification | Unique Peptides | Precursors | Spectra |
|---|---|---|---|---|
| K | Acetylation | 25 K | 27.2 K | 693 K |
| R | Methylation | 11.7 K | 13.5 K | 516 K |
| S | Phosphorylation | 26.9 K | 31.6 K | 1.1 M |
| T | Phosphorylation | 11.1 K | 12.2 K | 248 K |
| Y | Phosphorylation | 29.2 K | 36.1 K | 3.3 M |

Table 9 highlights interesting patterns observed in Figure 3. First, the same modification occurring at different residues can have varying effects on the peptide properties, implying that including amino acid PTM information is essential to achieve better predictions. Second, some modifications have the same Unimod ID and the same molecular structure but only differ in their stereo-chemistry (spatial arrangement of atoms), yet they impact the peptide properties differently. Such scenarios are present in modified sequences and require a proper representation of PTMs (via encoding and domain-specific features) to predict peptide properties accurately. Table 11 in Appendix Section D shows the impact of PTMs on retention time for the special cases from Table 9.

Table 8: Summary statistics of Test-PTM dataset.

| Residue | Modification | Unique Peptides | Precursors | Spectra |
|---|---|---|---|---|
| K | Acetylation | 198 | 237 | 47.2 K |
| K | Biotinylation | 197 | 225 | 25.7 K |
| K | Butyrylation | 200 | 241 | 47.5 K |
| K | Crotonylation | 200 | 237 | 48.5 K |
| K | Di-Methylation | 189 | 348 | 39.2 K |
| K | Formylation | 197 | 229 | 50.3 K |
| K | Glutarylation | 200 | 233 | 52.9 K |
| K | Ubiquitinylation | 200 | 382 | 52.4 K |
| K | Hydroxyisobutyrylation | 199 | 226 | 48.4 K |
| K | Malonylation | 198 | 224 | 35.1 K |
| K | Methylation | 194 | 365 | 45.2 K |
| K | Propionylation | 200 | 236 | 56.5 K |
| K | Succinylation | 197 | 233 | 46.9 K |
| K | Tri-Methylation | 186 | 329 | 38.1 K |
| P | Hydroxylation | 169 | 235 | 31.5 K |
| R | Citrullination | 184 | 247 | 37.9 K |
| R | Dimethyl-asymmetric | 181 | 313 | 38.3 K |
| R | Dimethyl-symmetric | 177 | 301 | 34.1 K |
| R | Methylation | 179 | 308 | 41.3 K |
| Y | Nitro | 175 | 215 | 58.5 K |
| Y | Phosphorylation | 174 | 217 | 101 K |
| N-Term | TMT | 6.2 K | 7.6 K | 32 K |
| K | TMT and Ubiquitinylation | 38.5 K | 51.7 K | 756 K |

Table 9: Examples of special amino acid-PTM pairs in our datasets.

| Modification (PTM) | Residue | Scenario |
|---|---|---|
| Phosphorylation | S / T / Y | Same PTM, different residue |
| Di-methylation | R / K | Same PTM, different residue |
| GlcNAC/GalNAC | S / T | Same PTM, different structure |
| Symmetric/Asymmetric di-methylation | R | Same PTM, different structure |

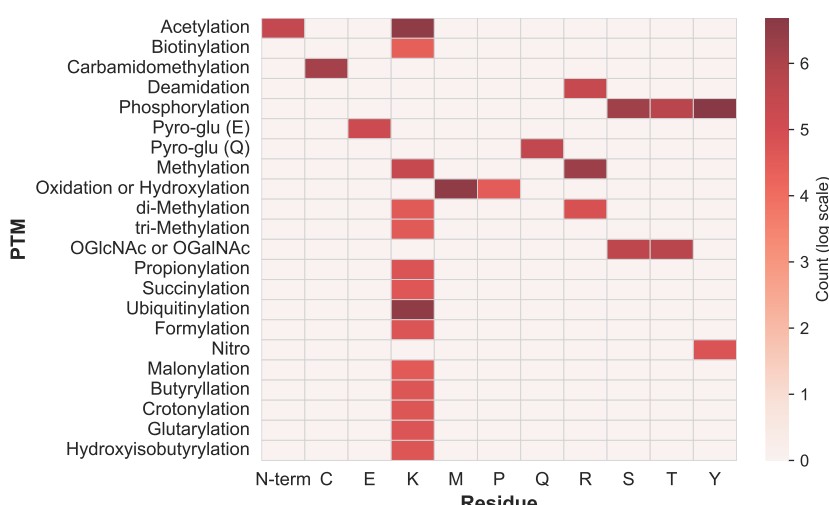

Figure 5: Heatmap indicating the frequency of each PTM occurring on different amino acids.

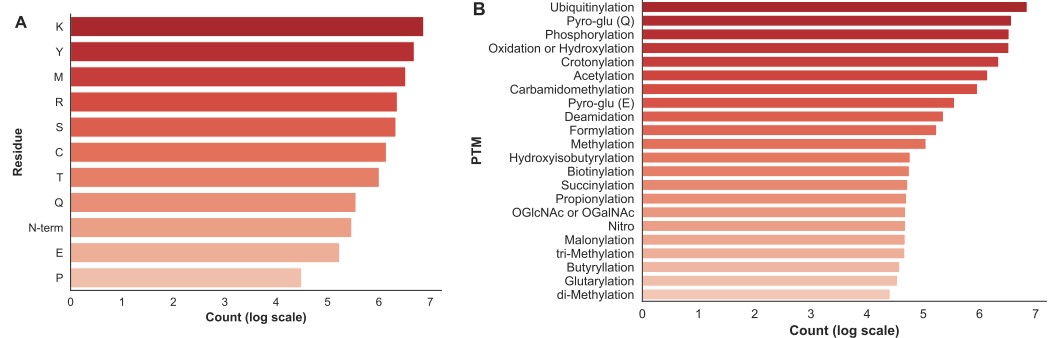

Figure 6: *A*: Frequency of each amino acid being reported as a modified site in the datasets. *B*: Frequency of occurrence of PTMs in the datasets.

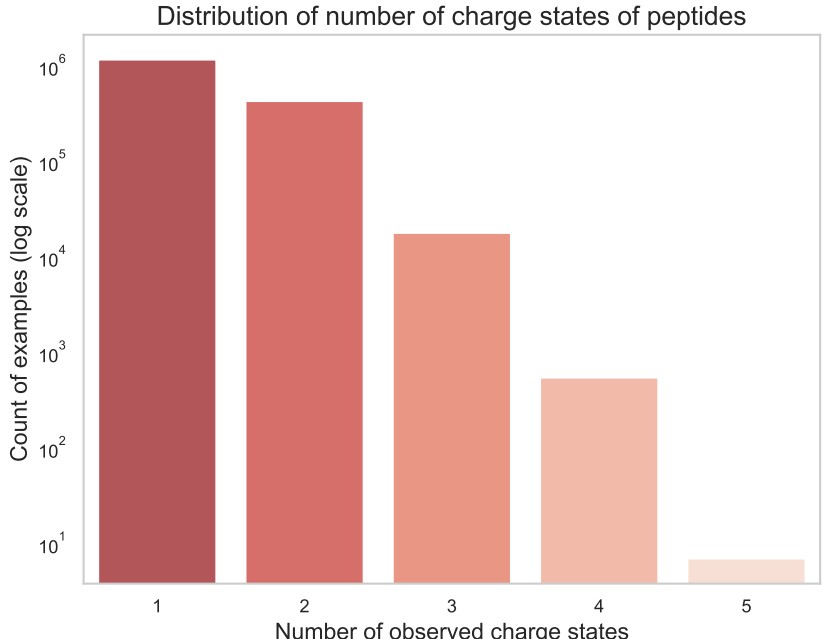

Figure 7: Bar plot of number of charge states for peptides in the datasets.

Table 10: Number of distinct charge states for peptides in the dataset.

| Number of Charge States | Number of Examples |
|---|---|
| 1 | 1,172,254 |
| 2 | 431,161 |
| 3 | 17,958 |
| 4 | 549 |
| 5 | 7 |

# D Evaluation and Metrics

For retention time prediction, the time delta at $95\%$ $\Delta t_{95\%}$ is the minimal time window containing the errors (residuals) between observed and predicted retention times for $95\%$ of the peptides [78].

The 95% threshold corresponds to $2\sigma$ of the residuals. This threshold can be increased to a higher percentage for stricter evaluation of model performance [21].

For intensities, the Spectral Angle is defined as follows for $V_a$ and $V_b$ being the observed and predicted intensity vectors [21]:

$$SA = 1 - \frac{2}{\pi}\cos^{-1}\left(\frac{V_a \cdot V_b}{\|V_a\| \cdot \|V_b\|}\right)$$

We provide code to compute the metrics in our data GitHub repository [62].

### D.1 Metrics for Datasets with PTMs

Throughout the paper, we calculated the Slope of a linear fit between two sets of iRT values. If the two sets of iRT values are perfectly aligned, we would expect a slope of 1 (e.g. model predictions against experimental values). We used this to highlight the impact of PTMs on retention time.

Although the two reported metrics (time delta 95 and Spectral angle) can be used for datasets with both unmodified and modified peptide sequences, some minor adaptations and granular evaluation can be conducted.

For example, when calculating the spectral angle, we align the peaks based on the annotation labels (y1, b1, y2, b2, etc...) to account for the m/z shift introduced by PTMs. Therefore, we only calculate how close the peak intensities are to each other.

For other tasks, reporting a suitable metric on subgroups of the dataset would give more insights into a model's performance. Potential subgroups include unmodified versus modified peptides, peptides with different modification types or peptides with modifications at different residues.

In Figure 8, we show iRT and intensity spectra behavior for different PTMs on TMT labeled peptides. We compare unmodified TMT-labeled peptides with modified TMT-labeled peptides. We also include the effect of TMT labeling on unmodified peptides. In Figure 9, we can see that PTMs with TMT show different behavior than PTMs on unlabeled peptides. Each point in the plot represents the effect of a single PTM, combined effects are not taken into account.

Tables 11, 12, 13, 14, 15, and 16 show the performance metrics reported on special cases and interesting subgroups of the datasets.

Table 11: Effect on retention time for the special residue-PTM pairs

| Residue | Modification | Slope | iRT95 | R2 |
|---------|--------------|-------|-------|------|
| S | Phosphorylation | 0.85 | 21.1 | 0.83 |
| T | Phosphorylation | 0.88 | 20.2 | 0.87 |
| Y | Phosphorylation | 0.85 | 23.4 | 0.81 |
| R | Di-Methylation | 0.91 | 8.7 | 0.97 |
| K | Di-Methylation | 0.98 | 15.2 | 0.94 |
| S | GlcNAC | 0.92 | 15.8 | 0.9 |
| S | GalNAC | 1.04 | 13.2 | 0.9 |
| T | GlcNAC | 0.96 | 16.8 | 0.88 |
| T | GalNAC | 1.05 | 15.9 | 0.9 |
| R | Symmetric Di-Methylation | 0.91 | 8.7 | 0.97 |
| R | aSymmetric Di-Methylation | 0.93 | 11.0 | 0.97 |

### D.2 Evaluation

Figure 10 shows the different distributions of the iRT residuals grouped by amino acid-PTM pair, sorted in ascending order by the delta iRT (difference between label and Prosit-DeltaAtoms prediction values for iRT). The results from Prosit model indicate that the model performs better than the DeepLC on most PTMs. Although there are some PTMs that both models struggle to predict, it shows that there is still room for improvement on the current SOTA models. An additional complexity can be observed on the Acetylated Lysine, showing a bi-modal distribution of delta iRT. Moreover, there is no consistent pattern among different PTMs, as they can shift the iRT to an earlier or a later point.

Table 12: Effect on retention time for the same PTM occurring at the same residue.

| Residue | Modification | Slope | iRT95 | R2 |
|---|---|---|---|---|
| K | Acetylation | 0.97 | 18.2 | 0.93 |
| K | Ubiquitinylation | 1.01 | 15.7 | 0.95 |
| K | Methylation | 1.1 | 18.8 | 0.94 |
| R | Citrullination | 0.85 | 16.2 | 0.97 |
| R | Methylation | 0.92 | 9.5 | 0.96 |
| Y | Phosphorylation | 0.84 | 24.2 | 0.8 |

Table 13: Effect on retention time for the same PTM occurring at a different residue.

| Residue | Modification | Slope | iRT95 | R2 |
|---|---|---|---|---|
| P | Hydroxylation | 0.94 | 9.9 | 0.97 |

Table 14: Effect on retention time for peptides with multiple PTMs

| Number of different PTMs | Slope | iRT95 | R2 |
|---|---|---|---|
| 1 | 0.93 | 18.7 | 0.95 |
| 2 | 0.89 | 20 | 0.95 |
| 3 | 0.88 | 20.5 | 0.94 |
| 4 | 0.86 | 22.6 | 0.92 |

Table 15: Effect on retention time for different PTMs

| Residue | Modification | Slope | iRT95 | R2 |
|---|---|---|---|---|
| K | Biotinylation | 1.19 | 38.1 | 0.94 |
| K | Butyrylation | 1.02 | 33.1 | 0.91 |
| K | Crotonylation | 1.04 | 34.4 | 0.91 |
| K | Di-Methylation | 0.98 | 15.2 | 0.94 |
| K | Formylation | 1.01 | 19 | 0.93 |
| K | Glutarylation | 1.07 | 21.5 | 0.93 |
| K | Hydroxyisobutyrylation | 0.96 | 24.9 | 0.93 |
| K | Malonylation | 0.94 | 20.7 | 0.93 |
| K | Propionylation | 0.99 | 23 | 0.92 |
| K | Succinylation | 1.05 | 22.7 | 0.94 |
| K | Tri-Methylation | 0.96 | 15.1 | 0.95 |
| R | Dimethyl-asymmetric | 0.93 | 11.1 | 0.97 |
| R | Dimethyl-symmetric | 0.91 | 8.7 | 0.97 |
| Y | Nitro | 0.91 | 29 | 0.84 |

Table 16: Effect on retention time for unmodified peptides.

| Residue | Modification | Slope | iRT95 | R2 |
|---|---|---|---|---|
| K | unmodified | 0.96 | 5.3 | 0.98 |
| R | unmodified | 0.95 | 4.4 | 0.97 |
| Y | unmodified | 0.96 | 6.2 | 0.97 |

Figure 11 shows the different distributions of the spectral angle grouped by amino acid-PTM pair, sorted in descending order by the median spectral angle with Prosit-DeltaAtoms predictions. A spectral angle below 0.6 usually indicates low similarity between the experimental and the predicted spectra, and hence, the predictions are not helpful in downstream tasks. Here, we divide the figure into three sub-plots to indicate which PTMs were used to train Prosit and which were used to train AlphapeptDeep. In Figure 11A, both models accurately predict the PTMs seen during

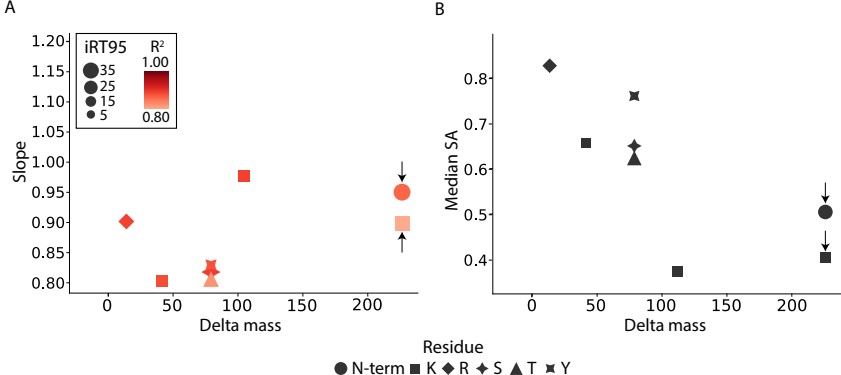

Figure 8: Scatter plots summarizing the iRT and fragment ion intensity difference between modified and matching unmodified TMT labeled peptides for all PTMs contained in the TMT dataset. Marked points show the difference between TMT labeled and unlabeled peptides.

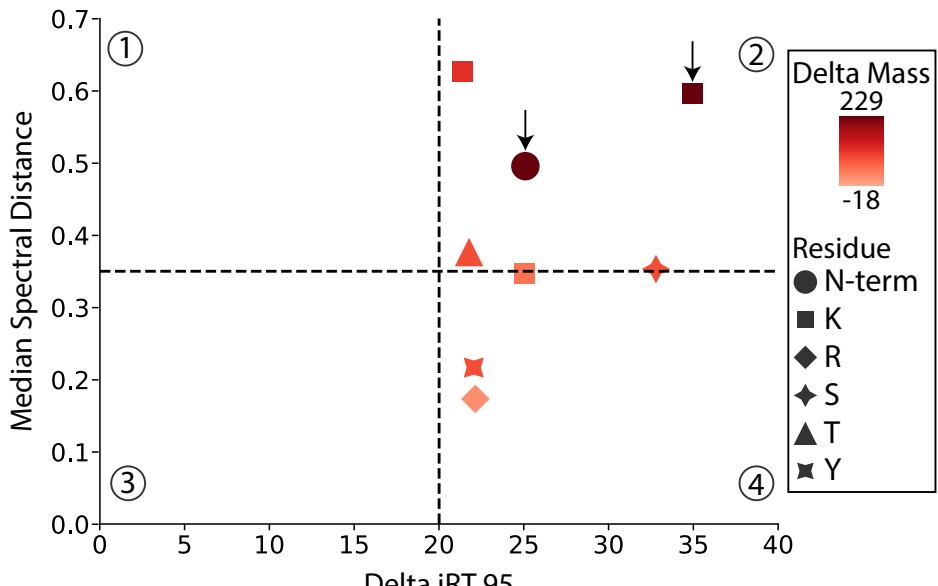

Figure 9: Impact of modifications on iRT vs. Intensity on TMT labeled peptides. Marked points show the difference between TMT labeled and unlabeled peptides.

training. In part B, AlphapeptDeep seems to be struggling with extrapolating on the unseen mods, while Prosit still performs well since these mods were included for the training phase. In Part C Prosit model generalizes to most of the unseen PTMs while still not getting the same performance as AlphapeptDeep. However, the model still struggles with some modifications: the first one is Biotinylation, and this might be because it has a bigger mass than most other PTMs and the model was not exposed to such modifications, the second is Malonylation, and this is mainly because, with this modification, there are very intense neutral loss peaks generated which changes the y- and b-ions intensity distribution entirely.

In Figure 12, Prosit shows comparable performance, while AlphapeptDeep seemingly struggles with this fragmentation method. The loss of performance in the AlphapeptDeep model is likely due to not having an input for the fragmentation method, and thus, it doesn't differentiate between HCD and CID fragmentation methods.

Figure 13 shows the different distributions of the precursor charge distributions grouped by amino acid-PTM pair, sorted in ascending order by the MAE (difference between label and Prosit prediction values for precursor charge distribution). We visualize how different modifications affect the charge

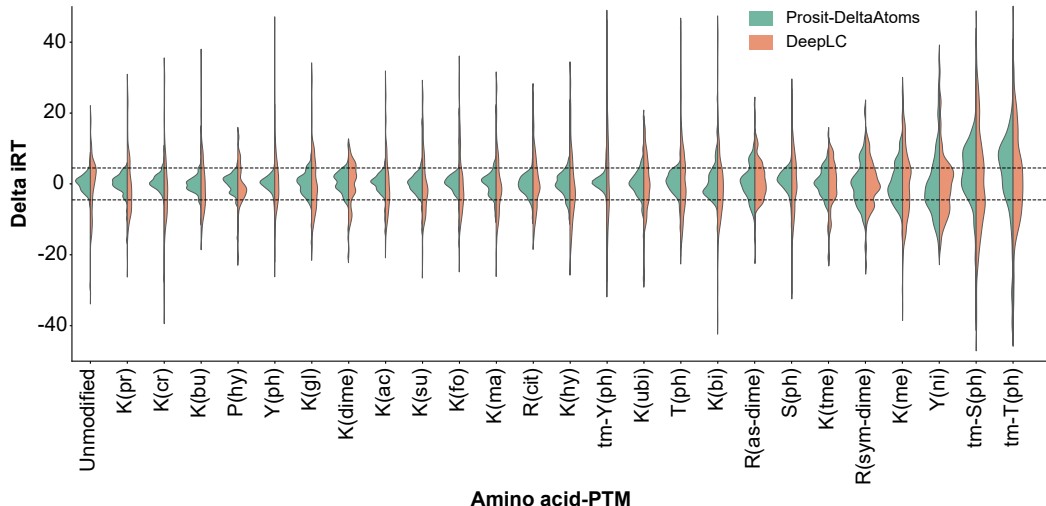

Figure 10: Retention Time prediction with Prosit and DeepLC violin plot with dashed lines indicating the delta iRT 95 for unmodified predictions with Prosit. Sequences are grouped by modifications.

distribution differently; some almost have no effect, and others have more prominent effects, leading to MAE values reaching 0.5.

# E    Elaboration on Supported Tasks

This dataset in combination with the original PROSPECT dataset [21] offers the opportunity to analyze in details the effect of PTMs on different peptide properties. While we covered in the main RT and fragment ion intensity, this can also be used for more such as precursor charge and neutral loss patterns [96, 97, 98]. It also provide the option of studying how the PTM location affects the peptide properties as well, while there are multiple algorithms trying to do this task [46, 47, 41], this is rather understudied with no proper study of the different effects of such change.

In Table 17, we provided a list of machine learning tasks that our datasets can primarily support in the context of PTMs. The datasets can be also used for several other tasks, either as-is or with the standard respective pre-processing required for the machine learning task at hand, without the need for deep domain knowledge or further annotations. Table 17 lists all tasksas feasible with our datasets to the best of our knowledge.

# F    Downstream Impact

Figure 14 shows that a model trained on PROSPECT PTM can lead to a gain of  20 % in PSMs and peptides after rescoring [108].

# G    More Details on Splitting the Datasets

Our general recommendation as described in Section 3.3 is to use Test-PTM as a hold-out dataset and split the three remaining datasets for training and validation of models. However, while iterating on model development and training, we additionally recommend splitting the three datasets into training, validation and test splits based on uniqueness of unmodified sequences (sequence-based disjoint split), where examples for the same unmodified peptide sequence should appear in only one of the splits [33]. Afterwards, the final selected model can be evaluated on the Test-PTM as a hold-out dataset.

For retention time, users of the dataset should filter out the quality control (QC) peptides that are used for RT calibration. This is detailed in [94].

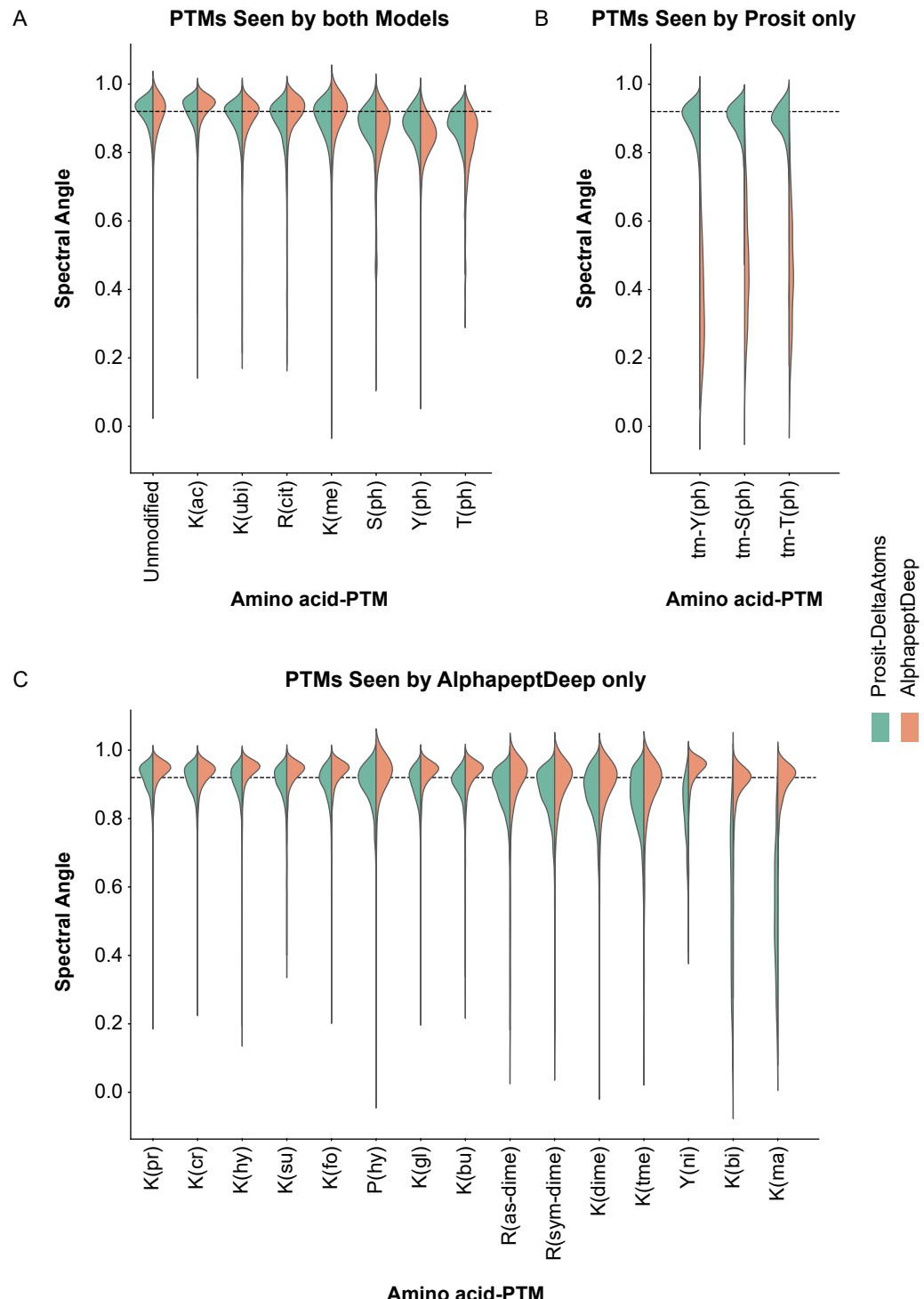

Figure 11: Violin plot with intensity predictions from Prosit and AlphaPeptDeep for HCD fragmentation. The dashed line indicating median SA for predictions on unmodified peptides with Prosit.

The counts in Table 18 show that the datasets contain mostly tryptic peptides. However, the TMT and the PTM datasets contain some non-tryptic peptides. Since studying PTMs in the context of tryptic peptides is more common, there should be no significant concerns about selection bias. Additionally,

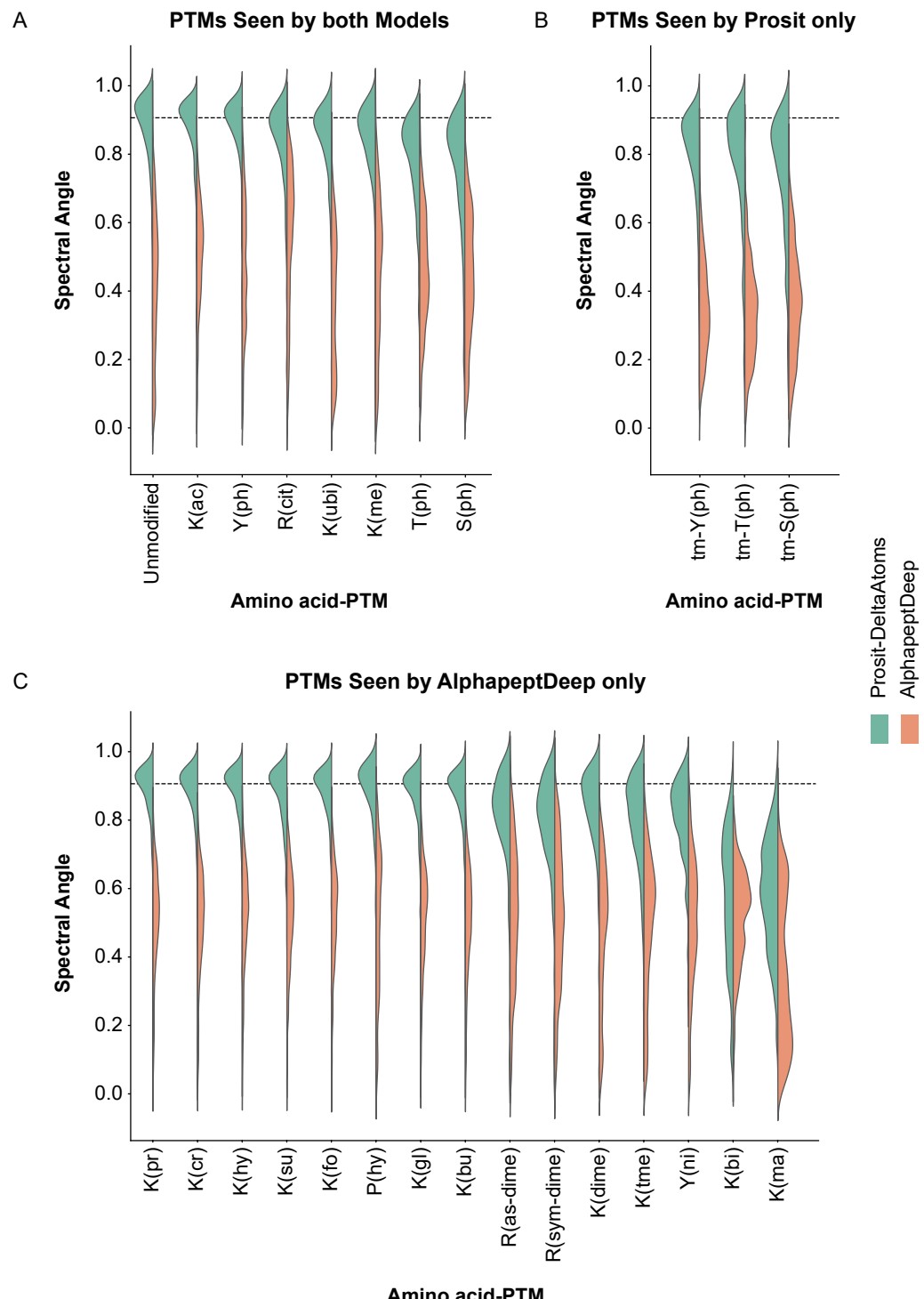

Figure 12: Violin plot with intensity predictions from Prosit and AlphaPeptDeep for CID fragmentation. The dashed line indicates the median SA for predictions on unmodified peptides with Prosit.

training models with the new datasets together with the unmodified peptides from PROSPECT [21] would help alleviate bias towards tryptic peptides.

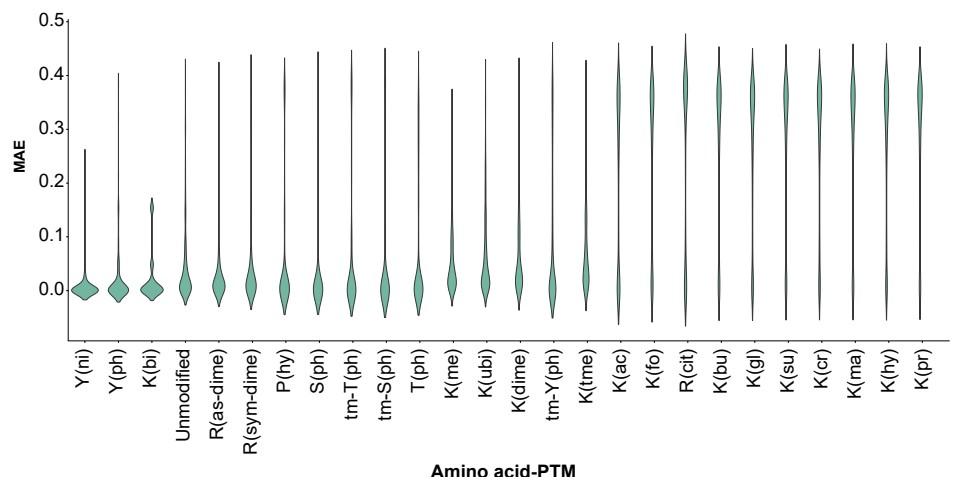

Figure 13: Distribution of predictions for precursor charge states.

Table 17: Listing of tasks feasible with PROSPECT PTM datasets.

| Name | Input | Target | Type | References |
|---|---|---|---|---|
| PTM site prediction | Sequence | PTM Site | Classification | [99, 100, 53] |
| Retention Time | Sequence | RT | Regression (single value) | [33, 38, 51, 40] |
| Retention Time with PTMs | Sequence | RT | Regression (single value) | [37] |
| Retention Time | Sequence | RT | Regression (distribution) | - |
| Intensity prediction | Sequence | intensities | Regression (vector) | [33, 34, 101] |
| Intensity prediction with PTMs | Sequence | intensities | Regression (vector) | [41, 39] |
| Fragment Presence | Sequence | Present/Not | Binary classification | [102] |
| Charge prediction | Sequence | Charge distribution | Regression | [69, 70] |
| De novo sequencing | Intensity | Sequence | Classification Ranking | [86, 87, 103, 104] |
| Sequence/Spectral Embedding | Sequence Spectra | Embeddings | Representation Similarity learning | [105, 84, 106] |
| Multiple properties | Sequence | RT charge intensity | Multi-task Learning | [69] |
| Sequence Clustering | Sequence | Cluster | Unsupervised Clustering | [107] |

## H   Experimental Details

For the evaluation of peptide sequences with PTMs, we trained various models. The first model is Prosit baseline [33]. The model was trained on the original PROSPECT dataset [21] for both

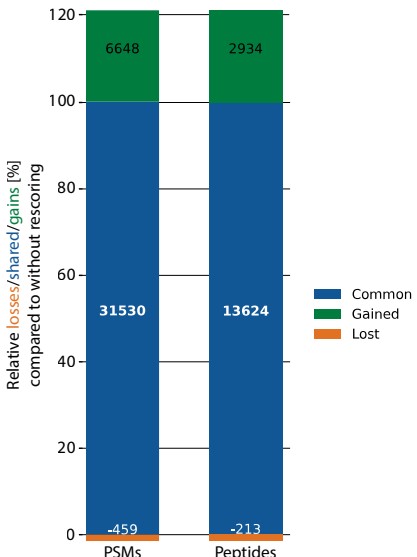

Figure 14: Stacked bar chart showing the number of confident PSMs (Peptide-Spectrum-Matches) (left) and peptides (right) below 1% FDR (False Discovery Rate) lost (orange), shared (blue) and gained (green) when rescoring using (Prosit) compared to the MSFragger [61] results.

Table 18: Count of Tryptic versus non-tryptic peptides.

| Dataset | Tryptic Peptides | Non-Tryptic Peptides | Tryptic Spectra | Non-Tryptic Spectra |
|---|---|---|---|---|
| TMT | 396 K | 318 K | 19 M | 9.2 M |
| PTM | 300 K | 7.2 K | 19.4 M | 267 K |
| TMT-PTM | 157 K | 1.6 K | 7.7 M | 52 K |

tasks, retention time prediction, and intensity prediction; this model serves as a baseline performance of how a model would predict different features without PTM encoding. Then, we trained other models using different ways of PTM encoding to see which might show the best performance. The Prosit naive model was trained on all datasets except Test-PTM. Hence, it does not support unseen PTMs but instead ignores them. Two variants of Prosit encoded PTMs with domain-specific features: Prosit-DeltaMass and Prosit-DeltaAtoms. Prosit-DeltaMass uses the mass introduced by the PTM as an input feature to the model, while Prosit-DeltaAtoms uses the atom count introduced by the PTM [40, 37]. AlphaPeptDeep [40] does not support N-term modifications.

The training and the inference were conducted using a single Nvidia A30 GPU. The Prosit model is an encoder-decoder RNN-based model that can predict indexed retention time and intensity spectrum for a given peptide sequence. The recurrent layers used in the architecture are Gated Recurrent Units (GRUs). More details on the architecture can be found in [33]. Training time is in the range of 2-3 hours on the unmodified dataset for retention time and 25 to 30 hours for MS/MS spectra.

For comparison, we ran inference on a DeepLC model for retention time prediction using an Nvidia A30 GPU. The DeepLC model architecture encodes input sequences and atom counts from amino acids and PTMs using 1D-convolutional and max pooling layers. The DeepLC model architecture includes a branch of fully-connected layers for global features engineered manually before training from the sequences [37]. Figure 15 depicts the difference between DeepLC predictions and the experimental iRT.

For MS spectrum prediction, we ran inference on AlphaPeptDeep [40] for MS spectra prediction using an Nvidia A30 GPU. The AlphaPeptDeep frameworks provides several model architectures. The pre-trained model we used encodes input sequences and atom counts from amino acids and PTMs using transformer layers with positional encoding.

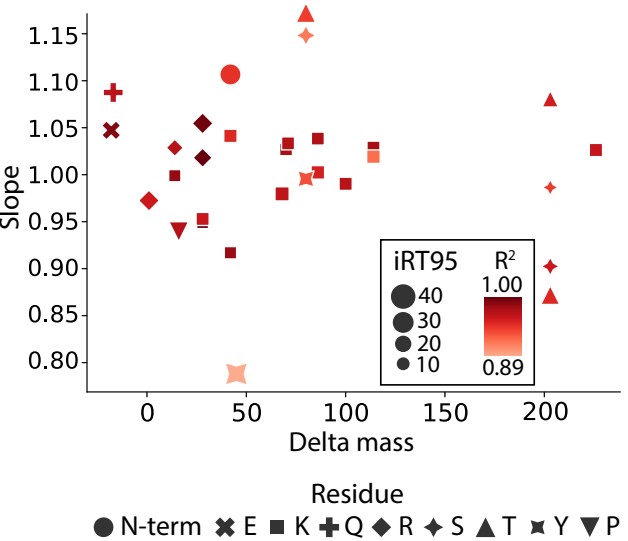

Figure 15: The iRT difference between DeepLC predictions on modified and matching unmodified peptides versus m/z values.

# I Dataset Documentation: Datasheet for Datasets

## I.1 Motivation

**For what purpose was the dataset created? Was there a specific task in mind? Was there a specific gap that needed to be filled? Please provide a description.**

The purpose is to introduce and establish multiple reference datasets for peptide sequences with different groups of Post-Translational Modifications present, complementing the original PROSPECT dataset, which contains unmodified peptide sequences. As of June 2024, the original PROSPECT dataset [28] was viewed on Zenodo 871 times and downloaded 1116 times.

The four new datasets are processed and curated for machine learning in Proteomics. Although they are not constrained to specific tasks, the focus is on three common tasks in proteomics; retention time prediction, MS/MS spectrum prediction, and precursor charge state prediction.

**Who created the dataset (e.g., which team, research group) and on behalf of which entity (e.g., company, institution, organization)?**

Computational Mass Spectrometry Chair at the School of Life Sciences, Technical University of Munich, Germany.

**Who funded the creation of the dataset? If there is an associated grant, please provide the name of the grantor and the grant name and number.**

The creation was partially funded by the following grants: European Proteomics Infrastructure Consortium providing access, Grant Number 823839 and Bundesministerium für Bildung und Forschung – BMBF, Grant Number 031L0008A.

**Other comments?**

We aim that this would be a start for different groups to release curated dataset with machine learning tasks in mind instead of only publishing raw datasets that required several processing steps before being useful for machine learning.

## I.2 Composition

**What do the instances that comprise the dataset represent (e.g., documents, photos, people, countries)? Are there multiple types of instances (e.g., movies, users, and ratings; people**

**and interactions between them; nodes and edges)? Please provide a description. How many instances are there in total (of each type, if appropriate)?**

Here we publish 4 different datasets. Instances of each dataset are peptide sequences, their corresponding annotations, and meta-data for spectra. We have in 4.6B unique peaks for 58.6M spectra of 1.2M unique peptides with 30 unique PTM-residue combinations. We uploaded 2 different file types; one for meta data for each spectrum and another with annotations.

**Does the dataset contain all possible instances or is it a sample (not necessarily random) of instances from a larger set? If the dataset is a sample, then what is the larger set? Is the sample representative of the larger set (e.g., geographic coverage)? If so, please describe how this representativeness was validated/verified. If it is not representative of the larger set, please describe why not (e.g., to cover a more diverse range of instances, because instances were withheld or unavailable).**

We have a dataset of all valid identifications from ProteomeTools PTMs and TMT raw data [48], one of the largest datasets with synthetic peptides.

**What data does each instance consist of? "Raw" data (e.g., unprocessed text or images) or features? In either case, please provide a description.**

We have the unprocessed meta data that we get from raw files generated as output from the mass spectrometer. We process the spectra from MS with the identifications that we get from MaxQuant [93] to annotate our dataset and annotate the fragment ions found in the spectra.

**Is there a label or target associated with each instance? If so, please provide a description.**

There are various machine learning problem formulations in proteomics, more details are in section 2. For the three tasks we focused on; namely, retention time, precursor charge state, and intensity prediction, the targets are retention time (and indexed retention time), precursor charge states, and the annotated spectra, respectively.

Instances of the dataset are generally linked together with the raw file and scan number associated with each spectra.

**Are relationships between individual instances made explicit (e.g., users' movie ratings, social network links)? If so, please describe how these relationships are made explicit.**

There are no direct relationships between different instances they might have some features in the metadata such as length, retention time, collision energy, PTM-residue combination and peptide sequence.

**Are there recommended data splits (e.g., training, development/validation, testing)? If so, please provide a description of these splits, explaining the rationale behind them.**

For MS Spectra, we recommend splitting data based on the peptide sequence so no peptide sequence is shared across different splits to avoid data leakage. Also splitting each pool in different files as each pool has different set of peptides to ensure that the splits has all the different types of peptides and didn't miss any. For charge state prediction, we recommend to split the data with a similar approach.

For retention time, while we suggest the same as Spectra in terms of not sharing sequences in different splits, we additionally recommend to filter examples and keep only one copy of each with the mean retention time of measurements for the same sequence. The mean retention time for each sequence can then be used for training the model.

The splits we used are available as ready-to-use datasets on the Hugging Face hub for the three tasks; retention time [63], MS2 [64], and charge state [65].

We suggest as well to use the PTM-Test as another holdout dataset for models trying to predict features for peptides with PTMs. We explain in details why this is useful in 3.3. More details are in Appendix Section G. The respective curated dataset on the Hugging Face hub is available in the three task repositories under a separate configuration denoted as *holdout*.

**Are there any errors, sources of noise, or redundancies in the dataset? If so, please provide a description.**

Our objective was to reduce the number of miss-identifications in the dataset, since we know which set of peptides we expect in each raw file, we remove all other identifications made by MaxQuant [93]. Although there might still be miss-identifications after this filtering, they would rather be less than 1%, which is the known acceptable cut off-in the field. There are redundancies as the same peptide would be measured multiple times but the spectra and Retention time would be slightly different in different measurements, we kept in this case all instances in the dataset.

**Is the dataset self-contained, or does it link to or otherwise rely on external resources (e.g., websites, tweets, other datasets)? If it links to or relies on external resources, a) are there guarantees that they will exist, and remain constant, over time; b) are there official archival versions of the complete dataset (i.e., including the external resources as they existed at the time the dataset was created); c) are there any restrictions (e.g., licenses, fees) associated with any of the external resources that might apply to a dataset consumer? Please provide descriptions of all external resources and any restrictions associated with them, as well as links or other access points, as appropriate.**

The dataset is self-contained as all the processed information is in one place. The only external resource is when users want to get access to the raw unprocessed data this is shared on pride archives [27, 25, 26]. All the archives are open access with a CC license and no restrictions on getting the data.

**Does the dataset contain data that might be considered confidential (e.g., data that is protected by legal privilege or by doctor– patient confidentiality, data that includes the content of individuals' non-public communications)? If so, please provide a description.**

No

**Does the dataset contain data that, if viewed directly, might be offensive, insulting, threatening, or might otherwise cause anxiety?**

No

## I.3   Collection

**How was the data associated with each instance acquired? Was the data directly observable (e.g., raw text, movie ratings), reported by subjects (e.g., survey responses), or indirectly inferred/derived from other data (e.g., part-of-speech tags, model-based guesses for age or language)? If the data was reported by subjects or indirectly inferred/derived from other data, was the data validated/verified? If so, please describe how.**

Raw files were acquired with a Mass spectrometer and peptide identifications were made with MaxQuant [93] (a software for database search). Here we depend on MaxQuant for identifications, but as mentioned in the previous section we remove miss-identifications to decrease the number of wrong labels. This is a particular strength of this dataset since we know which peptides exist per sample, based on the fact that they were specifically synthesized. For other datasets, we might only know to which organism they belong, which can lead to a higher number of misidentifications.

**What mechanisms or procedures were used to collect the data (e.g., hardware apparatuses or sensors, manual human curation, software programs, software APIs)? How were these mechanisms or procedures validated?**

Data was measured with Mass spectrometers and annotated with a software. Already explained in the question above how the dataset was validated.

**If the dataset is a sample from a larger set, what was the sampling strategy (e.g., deterministic, probabilistic with specific sampling probabilities)?**

No sampling was done, we include all peptides that were measured.

**Who was involved in the data collection process (e.g., students, crowdworkers, contractors) and how were they compensated (e.g., how much were crowdworkers paid)?**

Raw data were acquired by PhD students working on the ProteomeTools project [48] and annotated and curated by PhD students and the authors of the accompanying paper.

**Over what timeframe was the data collected? Does this timeframe match the creation timeframe of the data associated with the instances (e.g., recent crawl of old news articles)? If not, please describe the timeframe in which the data associated with the instances was created.**

Data acquisition started as early as 2017, but the time-frame doesn't affect the data in any shape or form.

**Were any ethical review processes conducted (e.g., by an institutional review board)? If so, please provide a description of these review processes, including the outcomes, as well as a link or other access point to any supporting documentation.**

No, the data is based on ProteomeTools which contains only synthetic peptide samples.

## I.4 Preprocessing/Cleaning/Labeling

**Was any preprocessing/cleaning/labeling of the data done (e.g., discretization or bucketing, tokenization, part-of-speech tagging, SIFT feature extraction, removal of instances, processing of missing values)? If so, please provide a description. If not, you may skip the remaining questions in this section.**

No.

**Was the "raw" data saved in addition to the preprocessed/cleaned/labeled data (e.g., to support unanticipated future uses)? If so, please provide a link or other access point to the "raw" data.**

The raw data is publicly available through the PRIDE archives [27, 25, 26]. The annotated datasets we provide are available on Zenodo [56, 57, 58, 59]. The processed and split task-specific datasets are available on the Hugging Face Hub [63, 64, 65].

**Is the software that was used to preprocess/clean/label the data available? If so, please provide a link or other access point.**

No, we used MaxQuant to remove misidentifications. Further processing of the datasets was done with Python scripts that are available in a dedicated dataset utilities GitHub repository [62].

## I.5 Usage

**Has the dataset been used for any tasks already? If so, please provide a description.**

Yes, parts of the dataset were previously used in different models for predicting fragment ions intensity and retention time, examples include the work in [43].

**Is there a repository that links to any or all papers or systems that use the dataset? If so, please provide a link or other access point.**

We referenced previous work in the paper and in Zenodo along with the dataset itself [57, 56, 58, 59].

**What (other) tasks could the dataset be used for?**

The data can be used for various tasks, examples include prediction of different peptide features, study double annotations for different peaks, assignment of annotations to peaks and localizing PTMs. For more details, please refer to Section 3 and Appendix Section E.

**Is there anything about the composition of the dataset or the way it was collected and preprocessed/cleaned/labeled that might impact future uses? For example, is there anything that a dataset consumer might need to know to avoid uses that could result in unfair treatment of individuals or groups (e.g., stereotyping, quality of service issues) or other risks or harms (e.g., legal risks, financial harms)? If so, please provide a description. Is there anything a dataset consumer could do to mitigate these risks or harms?**

No, not as far as we know.

**Are there tasks for which the dataset should not be used? If so, please provide a description.**

No, not as far as we know.

### I.6 Distribution

**Will the dataset be distributed to third parties outside of the entity (e.g., company, institution, organization) on behalf of which the dataset was created? If so, please provide a description.**

Both the raw data from ProteomeTools and our dataset PROSPECT are publicly available. Every third party outside the entity on behalf of which the dataset was generated has access to it now.

**How will the dataset will be distributed (e.g., tarball on website, API, GitHub)? Does the dataset have a digital object identifier (DOI)?**

We uploaded the 4 datasets to Zenodo, each with a dedicated DOI [57, 56, 58, 59]. The processed and split task-specific datasets are uploaded to the Hugging Face Hub with dedicated DOIs [63, 64, 65]. We also have a GitHub repository with utilities to download the dataset [62].

**When will the dataset be distributed?**

We published the datasets on Zenodo in October 2023 and June 2024. We published the accessible splitted, aggregated, and processed datasets to the Hugging Face Hub in June 2024.

**Will the dataset be distributed under a copyright or other intellectual property (IP) license, and/or under applicable terms of use (ToU)? If so, please describe this license and/or ToU, and provide a link or other access point to, or otherwise reproduce, any relevant licensing terms or ToU, as well as any fees associated with these restrictions.**

Open access, Creative Commons Attributions 4.0 International.

**Have any third parties imposed IP-based or other restrictions on the data associated with the instances? If so, please describe these restrictions, and provide a link or other access point to, or otherwise reproduce, any relevant licensing terms, as well as any fees associated with these restrictions.**

No.

**Do any export controls or other regulatory restrictions apply to the dataset or to individual instances? If so, please describe these restrictions, and provide a link or other access point to, or otherwise reproduce, any supporting documentation.**

No.

### I.7 Maintenance

**Who will be supporting/hosting/maintaining the dataset?**

Professorship for Computational Mass Spectrometry at the Technical University of Munich (TUM). The datasets are hosted on Zenodo [56, 57, 58, 59] and the task-specific processed datasets are hosted on the Hugging Face [63, 64, 65]. Current maintainers are the authors and later other members of the Professorship at TUM.

**How can the owner/curator/manager of the dataset be contacted (e.g., email address)?**

Mathias Wilhelm (mathias.wilhelm@tum.de).

**Is there an erratum? If so, please provide a link or other access point.** No.

**Will the dataset be updated (e.g., to correct labeling errors, add new instances, delete instances)? If so, please describe how often, by whom, and how updates will be communicated to dataset consumers (e.g., mailing list, GitHub)?**

If we detect further misidentifications or improve the quality of annotations, we will release subsequent versions with the respective updates. This will be versioned and announced in Zenodo, the Hugging Face Hub, and GitHub.

**If the dataset relates to people, are there applicable limits on the retention of the data associated with the instances (e.g., were the individuals in question told that their data would be retained for a fixed period of time and then deleted)? If so, please describe these limits and explain how they will be enforced.**

No, the dataset is neither related to people nor based on human samples.

**Will older versions of the dataset continue to be supported/hosted/maintained? If so, please describe how. If not, please describe how its obsolescence will be communicated to dataset consumers.**

Yes, different versions will be maintained on Zenodo and the Hugging Face Hub.

**If others want to extend/augment/build on/contribute to the dataset, is there a mechanism for them to do so? If so, please provide a description. Will these contributions be validated/verified? If so, please describe how. If not, why not? Is there a process for communicating/distributing these contributions to dataset consumers? If so, please provide a description.**

We welcome and encourage others to extend/augment/build on/contribute to the dataset. We suggested initiating contact with our professorship and we will discuss the best options for communicating/distribution the additions.

## J  Author Statement

The authors confirm all responsibility in case of violation of rights and confirm the licence associated with the dataset.

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
