# OpenReview forum: "PROSPECT PTMs: Rich Labeled Tandem Mass Spectrometry Dataset of Modified Peptides for Machine Learning in Proteomics"
_NeurIPS.cc/2024/Datasets_and_Benchmarks_Track — NeurIPS 2024 Track Datasets and Benchmarks Poster_

### Official Review · Reviewer_LgtK · 2024-07-02

**Rating:** 5
**Confidence:** 5
**Correctness:** Yes.
**Clarity:** Yes.

**Review:**

This work exhibits high quality through its meticulous assembly of four extensive datasets comprising over a million unique modified peptide sequences, demonstrating a robust annotation and validation process. The clarity of the manuscript is commendable, with detailed explanations of dataset generation, encoding techniques, and potential applications, ensuring comprehensibility for both novice and expert readers. The significance of this contribution is substantial, as it paves the way for more accurate and comprehensive machine learning models in proteomics, potentially enabling better prediction of peptide properties and protein functions.

**Strengths:**

1. This paper is well-organized and clearly written.
2. The datasets’ comprehensiveness, with 1.2 million unique modified peptide sequences, addresses a critical gap in existing proteomics research.
3. The detailed annotation and robust validation process enhance the datasets’ reliability.

**Additional Feedback:**

N/A

**Documentation:**

Yes.

**Limitations:**

1. The reliance on synthetic data might not fully capture the complexity of natural proteomes, possibly leading to less accurate predictions in real-world applications.
2. The authors should offer user-friendly tools or code to assist researchers in utilizing the provided datasets.

**Opportunities For Improvement:**

The reliance on synthetic data from ProteomeTools may limit the generalizability to natural biological samples. Additionally, while the datasets facilitate various downstream tasks, this work could further benefit from more extensive benchmarking against diverse models to fully demonstrate their impact.

**Relation To Prior Work:**

Yes.

**Summary And Contributions:**

This paper introduces a comprehensive dataset aimed at enhancing machine learning applications in proteomics. The authors present four high-quality labeled datasets containing 1.2 million unique modified peptide sequences and 30 distinct amino acid-PTM pairs. This work addresses the current lack of high-quality reference data for modified peptides, which has hindered the development of machine learning models for predicting MS/MS-related properties. The datasets support various downstream tasks, such as retention time, fragment ion intensity, and precursor charge prediction, and facilitate the encoding of PTMs into model inputs, improving the accuracy and robustness of predictions involving modified peptides. The paper demonstrates the datasets’ utility through benchmarking results, highlighting their significance in advancing proteomics research and clinical applications.

---

> ### Author Rebuttal · Authors · 2024-08-16
>
> We thank the reviewer for highlighting areas where we can improve our work and providing valuable feedback. We want to address the comments by providing some clarifications and mentioning the changes we will implement in the final revision of the paper.
>
> ----
>
> **Synthetic data and generalizability**
>
> In ProteomeTools [1], almost all peptides selected for synthesis were previously observed in biological samples, and then systematically measured using tandem mass-spectrometry. Therefore, although the peptides are synthetic, the data is not artificial and generalizes certainly to the human proteome - and by extension to any other organism, since a mass spectrometer does not differentiate the origin/source of a peptide. In fact, a lot of peptides observed in human are shared with other organisms. Additionally, the choice of using synthesized peptides has many advantages [1]: (1) knowledge of the peptides present in the sample leads to fewer false-positive, rendering it perfect for machine learning; (2) synthetic peptides are not impacted by varying protein abundance in a biological sample, leading to overall higher spectral quality, (3) less chimercity (less cases where multiple peptides are co-isolated), which reduces the noise and the false-positive annotations, and (4) enables the systematic acquisition of various fragmentation settings, reducing redundancy, and increasing consistency. From our experiments, and as observed by the widespread use of Prosit, models trained on this data show great generalizability to other organisms and achieved SOTA performance on unseen data [2,3,4,5].
>
> ----
>
> **Downstream impact**
>
> To further demonstrate the impact of the PROSPECT PTMs dataset, we have added an explicit example of a downstream task, showing the improvement of PSM and peptide identification rates on a TMTpro (18plex) phosphoproteome dataset [11]. We chose this dataset because PROSPECT PTMs dataset does not contain TMTpro labeled peptides and to showcase the multi-PTM aspect of trained models. Figure 1 in the attached PDF shows that a model trained on PROSPECT PTM can lead to a gain of ~20% in PSMs and peptides after rescoring.
> Besides Prosit, many other peptide property prediction models, which also generalize well on downstream tasks,  were trained or evaluated on the ProteomeTools data, like pDeep [10], AlphaPeptDeep [11], SpecEncoder [12] or InstaNovo [13], underlining the importance of PROSPECT and the proposed extension to PTMs.
> For benchmarking, we picked the most widely used and trusted models. Because of the lack of datasets like PROSPECT PTM, most models only explicitly support a small number of PTMs, and thus the benchmarking performed here is provided to give some indication of research gap. On the example of fragment ion intensity prediction, while some models claim to support more PTMs, this is often only achieved by shifting the respective fragment ions in m/z space, not really addressing the impact of the PTM on intensities. On the example of precursor charge state prediction, currently no model is capable of predicting more than very few predefined PTMs.
> Further, we believe PROSPECT PTMs dataset is not limited to applications for boosting identifications. Peptide property predictions find applications in spectral library generation that would greatly benefit from additional predictors to reduce the library size, such as precursor charge state. This is critical, for example, in metaproteomics experiments where analysis suffers from huge search space [12]. A smaller library size is aimed to increase sensitivity and specificity while reducing the computational cost. Peptide property predictions can also be used for single peptides, e.g. in targeted proteomics experiments, to pick the best collision energies [13], or to validate single peptide identifications by (visually) comparing experimental vs. predicted fragmentation spectra [19], e.g. in immunopeptidomics for neo-antigen validation, all relevant for translating findings to medicine [14].
> We will clarify this in the revision of the paper.
>
> ----
>
> **Tools to utilize the data and facilitate its use**
>
> We provide the following approaches to enable accessibility to the datasets, as shown in Figure 1 in the Appendix (also in Figure 3 in the attached PDF with improvements for clarity):
> a ready-to-use dataset collection hosted on the Hugging Face Hub [6] that includes a preprocessed and split dataset for each of the three tasks we discuss in the manuscript.
> Zenodo datasets to have access to the full annotated data and an accompanying GitHub repository [7] with scripts to easily download, filter, and analyze the datasets.
>
> We opted for Hugging Face Hub since it is the machine learning community standard for sharing datasets and making them accessible. For the large annotated data, we chose Zenodo since it is an open repository that is commonly used to host large research datasets. Since Zenodo does not provide a programmatic way to download the data, we provided the GitHub repository [7] to facilitate downloading the data. With these resources available for users, we facilitate using the datasets for training, evaluating, and running pre-trained models (using tools such as DLOmix[10], Koina[8]) and for downstream tasks (using a pipeline such as Oktoberfest [9]).
>
> We will highlight this more clearly in the text and possibly move the Figure to the main manuscript.
>
>
> Thanks again for your valuable feedback, and we appreciate any further elaboration or feedback.
>
> Note: references will be in a separate comment.

---

> > ### Author Response · Authors · 2024-08-16
> > **References for Rebuttal**
> >
> > [1] Zolg, D. P., Wilhelm, M., Schnatbaum, K., Zerweck, J., Knaute, T., Delanghe, B., ... & Kuster, B. (2017). Building ProteomeTools based on a complete synthetic human proteome. Nature methods, 14(3), 259-262.
> >
> > [2] Gessulat, S., Schmidt, T., Zolg, D. P., Samaras, P., Schnatbaum, K., Zerweck, J., ... & Wilhelm, M. (2019). Prosit: proteome-wide prediction of peptide tandem mass spectra by deep learning. Nature methods, 16(6), 509-518.
> >
> > [3] Gabriel, W., The, M., Zolg, D. P., Bayer, F. P., Shouman, O., Lautenbacher, L., ... & Wilhelm, M. (2022). Prosit-TMT: deep learning boosts identification of TMT-labeled peptides. Analytical Chemistry, 94(20), 7181-7190.
> >
> > [4] Wilhelm, M., Zolg, D. P., Graber, M., Gessulat, S., Schmidt, T., Schnatbaum, K., ... & Kuster, B. (2021). Deep learning boosts sensitivity of mass spectrometry-based immunopeptidomics. Nature communications, 12(1), 3346.
> >
> > [5] Adams, C., Gabriel, W., Laukens, K., Picciani, M., Wilhelm, M., Bittremieux, W., & Boonen, K. (2024). Fragment ion intensity prediction improves the identification rate of non-tryptic peptides in timsTOF. Nature communications, 15(1), 3956.
> >
> > [6] HuggingFace PROSPECT PTMs Dataset collection: ​​https://huggingface.co/collections/Wilhelmlab/prospect-ptms-665db48431a7e844634660ba
> >
> > [7] GitHub PROSPECT PTMs: https://github.com/wilhelm-lab/PROSPECT
> >
> > [8] Lautenbacher, L., Yang, K., Kockmann, T., Panse, C., Chambers, M., Kahl, E., ... & Wilhelm, M. (2024). Koina: Democratizing machine learning for proteomics research. bioRxiv, 2024-06.
> >
> > [9] Picciani, M., Gabriel, W., Giurcoiu, V. G., Shouman, O., Hamood, F., Lautenbacher, L., ... & Wilhelm, M. (2024). Oktoberfest: Open‐source spectral library generation and rescoring pipeline based on Prosit. Proteomics, 24(8), 2300112.
> >
> > [10] GitHub DLOmix: Python framework for Deep Learning in Proteomics https://github.com/wilhelm-lab/dlomix
> >
> > [11] Li J., Cai Z., Bomgarden R. D., Pike I., Kuhn K., Rogers J. C., Roberts T. M., Gygi S. P. & Paulo J. A. (2021). TMTpro-18plex: The Expanded and Complete Set of TMTpro Reagents for Sample Multiplexing. J Proteome Res., 20(5), 2964–2972
> >
> > [12] Rechenberger J., Samaras P., Jarzab A., Behr J., Frejno M., Djukovic A., Sanz J., González-Barberá E. M., Salavert M., López-Hontangas J. L., Xavier K. B., Debrauwer L., Rolain J.-M., Sanz M., Garcia-Garcera M., Wilhelm M., Ubeda C. & Kuster B. (2019). Challenges in Clinical Metaproteomics Highlighted by the Analysis of Acute Leukemia Patients with Gut Colonization by Multidrug-Resistant Enterobacteriaceae. Proteomes, 7, 2.
> >
> > [13] Révész Á., Hevér H., Steckel A., Schlosser G., Szabó D., Vékey K., Drahos L. (2021). Collision energies: Optimization strategies for bottom‐up proteomics. Mass Spec Rev., 42, 1261–1299
> >
> > [14] Tretter C., … & Krackhardt A. M. (2023). Proteogenomic analysis reveals RNA as a source for tumor-agnostic neoantigen identification. Nature communications, 14, 4632
> >
> > [15] Zhou X.-X., Zeng W.-F., Chi H., Luo C., Liu C., Zhan J., He S.-M. & Zhang Z. (2017). pDeep: Predicting MS/MS Spectra of Peptides with Deep Learning. Anal. Chem., 89, 12690−12697
> >
> > [16] Zeng W.-F., Zhou X.-X., Willems S., Ammar C., Wahle M.,Bludau I., Voytik E., Strauss M. T. & Mann M. (2022). AlphaPeptDeep: a modular deep learning framework to predict peptide properties for proteomics. Nature communications, 13, 7238
> >
> > [17] Liu K., Tao C., Ye Y. & Tang H. (2024). SpecEncoder: deep metric learning for accurate peptide identification in proteomics. Bioinformatics, 40, i257–i265
> >
> > [18] Eloff, K.; Kalogeropoulos, K.; Morell, O.; Mabona, A.; Jespersen, J. B.; Williams, W.; Beljouw, S. P. B.; van Skwark, M.; Laustsen, A. H.; Brouns, S. J. J.; Ljungars, A.; Schoof, E. M.; Goey, J. V.; Keller, U. auf dem; Beguir, K.; Carranza, N. L.; Jenkins, T. P. (2023). De Novo Peptide Sequencing with InstaNovo: Accurate, Database-Free Peptide Identification for Large Scale Proteomics Experiments. bioRxiv,  DOI: 10.1101/2023.08.30.555055
> >
> > [19] Schmidt T., Samaras P., Dorfer V., Panse C., Kockmann T., Bichmann L., van Puyvelde B., Perez-Riverol Y.,  Deutsch E. W., Kuster B. & Wilhelm M. (2021). Universal Spectrum Explorer: A Standalone (Web-)Application for Cross-Resource Spectrum Comparison. J. Proteome Res., 20, 3388−3394

---

### Official Review · Reviewer_9C1C · 2024-07-21
**An important problem area in computational biology**

**Rating:** 7
**Confidence:** 5

**Review:**

See following sections.

**Strengths:**

Post-translational modifications are an important area of biology, certain variants of which are challenging to measure experimentally. Large-scale annotated datasets for these are lacking, which focus largely on unmodified peptide sequences. This work will facilitate machine learning approaches to detecting these.

While it is in my opinion self-evident that incorporating PTM information is necessary to predict properties of PTM'ed peptides, the authors nonetheless make the point convincingly through their evaluations.

The supplement is extremely thorough.

**Additional Feedback:**

I reviewed this submission last year and am overall pleased with the improvements made.

**Clarity:**

The abstract could be shortened. Overall, the language is pretty dense. The authors do an acceptable job of explaining jargon in this area to potential ML readers.

What is the "novel unpublished data" on line 87? Is this experimental MS data, or annotations?

On line 212, clarify that the annotations are generated by an "expert system" in the classical AI sense, as opposed to manual annotation by human experts.

In Figure 2a: please explain what slope, iRT95, and R2 are, or refer the reader in that caption to the relevant portion of the text. In particular, the slope in question is unclear until the supplement is consulted.

The mathematical definitions of "slope" and "spectral angle" should be moved from the supplement to the main body.

As stated previously, the variant of mass spectrometer used to acquire these data should be stated.

**Correctness:**

The dataset from which these spectra are derived, ProteomeTools, is widely used in the proteomics community and considered of high quality. The authors' annotation protocol follows standard practice in this field.

The splitting protocol is sensible. To get a sense of how similar the peptide sequences are across splits, and alleviate any potential concerns on the part of the reader about train-test leakage, it would be helpful if the authors indicated histograms of e.g. Levenshtein edit distances between peptides across splits.

**Documentation:**

A thorough datasheet-for-datasets is provided. Dataset is accessible in Parquet format via HuggingFace.

**Ethics:**

No concerns.

**Limitations:**

The authors should point out that ML models trained on their dataset are likely to inherit whatever bias is present in the expert system used to annotate them.

What sort of mass spectrometer was used to acquire the data? (e.g. Orbitrap, QTOF) - this also presents a source of bias and should be indicated.

**Opportunities For Improvement:**

The evaluation is acceptable, but its conclusions are also somewhat obvious - it would be surprising indeed if PTMs did not affect these properties. I would argue that these predictions of peptide MS/MS properties are really only of interest insofar as they lead to improved peptide identification rates on real data, and it would have been nice to see that explicitly.

**Relation To Prior Work:**

Paper is an extension of a previous contribution by the authors, PROSPECT, that did not include post-translationally modified peptides.

**Summary And Contributions:**

The authors provide a dataset, PROSPECT-PTM, comprising post-translationally modified peptides, their fragmentation mass spectra, retention times, and charge state distributions.

---

> ### Author Rebuttal · Authors · 2024-08-16
>
> We thank the reviewer for highlighting areas where we can improve our work and providing valuable feedback. We want to address the comments by providing some clarifications and mentioning the changes we will implement in the final revision of the paper.
>
> -----
>
> **Obvious conclusions on PTMs impact**
>
> While the general conclusion that PTMs affect peptide properties is expected, the effects vary across different properties and PTM structures. We demonstrate this variation for retention time and intensity in Figure 5 in the Appendix (also in the attached PDF with improvements for clarity). Here, we highlight the impact of PTMs on two properties, each axis shows the change in the respective property quantified by a suitable metric (delta-95 for retention time and spectral distance for fragment ion intensity), comparing the raw values of modified and unmodified versions of the same peptide. Depending on the quadrant where a PTM lies, this indicates how much it impacts the two properties in question (i.e. PTMs that do not change either RT or intensity, only either, or both). As shown in the Figure, all quadrants have PTMs, indicating the varying impact on the properties.
> We will move this figure to the main manuscript in the final revision, improve its overall readability, and elaborate on this in the text.
>
> -----
>
> **Predictions only of interest insofar as they lead to improved peptide identification**
>
> As rightfully pointed out in the review, we have added an explicit example of a downstream task, demonstrating the improvement of PSM and peptide identification rates on a TMTpro (18plex) phosphoproteome dataset [1]. We chose this dataset because PROSPECT PTMs dataset does not contain TMTpro labeled peptides and to showcase the multi-PTM aspect of trained models. Figure 1 in the attached PDF shows that a model trained on PROSPECT PTMs can lead to a gain of ~20% in PSMs and peptides after rescoring.
>
> Besides Prosit, many other peptide property prediction models, which also generalize well on downstream tasks,  were trained or evaluated on the ProteomeTools data, like pDeep [10], AlphaPeptDeep [11], SpecEncoder [12] or InstaNovo [13], underlining the importance of PROSPECT and the proposed extension to PTMs.
> Further, we believe PROSPECT PTMs dataset is not limited to applications for boosting identifications. Peptide property predictions find applications in spectral library generation that would greatly benefit from additional predictors to reduce the library size, such as PTM-sensitive precursor charge state. This is critical, for example, in metaproteomics experiments where analysis suffers from huge search space [7]. A smaller library size is aimed to increase sensitivity and specificity while reducing the computational cost. Peptide property predictions can also be used for single peptides, e.g. in targeted proteomics experiments, to pick the best collision energies [8], or to validate single peptide identifications by (visually) comparing experimental vs. predicted fragmentation spectra [14], e.g. in immunopeptidomics for neo-antigen validation, all relevant for translating findings to medicine [9].
> We will clarify this in the revision of the paper.
>
> ----
>
> **Data bias due to expert system annotations and mass spectrometer used**
>
> These are good points to highlight, thank you. We will add a sentence to the limitations and highlight the bias.
> To show the dataset's utility, we trained and evaluated our models on a selected subset of the fragment ions (b and y ions without neutral losses), which covers most of the intensity in the experimental spectrum. The models achieved a performance that is useful for downstream tasks as reported in the Evaluation section, in published research [2,3,4,5], and as we show in the attached PDF in Figure 1 for an example downstream rescoring task. While predicting y/b ions is one possible formulation of the fragment intensity task, our datasets enable various other formulations, including predicting other ions or even the full experimental spectrum.
>
> For the mass spectrometer, we will elaborate on this in the text to mention the limitations of the data. Although the mass analyzers used to acquire the datasets are Orbitrap and Iontrap, our experiments show that models trained on this dataset (e.g. Prosit), generalized reasonably well to e.g. TOFs, leading to similar increases in peptide identifications when using such models for rescoring [5].
>
> Also, since the physical peptides used to generate the reference data are still available, we expect further data from other peptide sets (e.g. PTMs) and mass spectrometers (e.g. Waters) to be added over the next years, further reducing biases and covering additional experimental settings. To show this, we have added a recently acquired timsTOF fragment ion intensity dataset to HuggingFace [6]. This dataset was acquired using the same physical peptides as the Orbitrap and Iontrap data.
>
> Thanks again for your valuable review, and we appreciate any further elaboration or feedback.
>
> Note: references will be in a separate comment.

---

> > ### Author Response · Authors · 2024-08-16
> > **References for Rebuttal**
> >
> > [1] Li J., Cai Z., Bomgarden R. D., Pike I., Kuhn K., Rogers J. C., Roberts T. M., Gygi S. P. & Paulo J. A. (2021). TMTpro-18plex: The Expanded and Complete Set of TMTpro Reagents for Sample Multiplexing. J Proteome Res., 20(5), 2964–2972
> >
> > [2] Gessulat, S., Schmidt, T., Zolg, D. P., Samaras, P., Schnatbaum, K., Zerweck, J., ... & Wilhelm, M. (2019). Prosit: proteome-wide prediction of peptide tandem mass spectra by deep learning. Nature methods, 16(6), 509-518.
> >
> > [3] Gabriel, W., The, M., Zolg, D. P., Bayer, F. P., Shouman, O., Lautenbacher, L., ... & Wilhelm, M. (2022). Prosit-TMT: deep learning boosts identification of TMT-labeled peptides. Analytical Chemistry, 94(20), 7181-7190.
> >
> > [4] Wilhelm, M., Zolg, D. P., Graber, M., Gessulat, S., Schmidt, T., Schnatbaum, K., ... & Kuster, B. (2021). Deep learning boosts sensitivity of mass spectrometry-based immunopeptidomics. Nature communications, 12(1), 3346.
> >
> > [5] Adams, C., Gabriel, W., Laukens, K., Picciani, M., Wilhelm, M., Bittremieux, W., & Boonen, K. (2024). Fragment ion intensity prediction improves the identification rate of non-tryptic peptides in timsTOF. Nature communications, 15(1), 3956.
> >
> > [6] https://huggingface.co/datasets/Wilhelmlab/timstof-ms2
> >
> > [7] Rechenberger J., Samaras P., Jarzab A., Behr J., Frejno M., Djukovic A., Sanz J., González-Barberá E. M., Salavert M., López-Hontangas J. L., Xavier K. B., Debrauwer L., Rolain J.-M., Sanz M., Garcia-Garcera M., Wilhelm M., Ubeda C. & Kuster B. (2019). Challenges in Clinical Metaproteomics Highlighted by the Analysis of Acute Leukemia Patients with Gut Colonization by Multidrug-Resistant Enterobacteriaceae. Proteomes, 7, 2.
> >
> > [8] Révész Á., Hevér H., Steckel A., Schlosser G., Szabó D., Vékey K., Drahos L. (2021). Collision energies: Optimization strategies for bottom‐up proteomics. Mass Spec Rev., 42, 1261–1299
> >
> > [9] Tretter C., … & Krackhardt A. M. (2023). Proteogenomic analysis reveals RNA as a source for tumor-agnostic neoantigen identification. Nature communications, 14, 4632
> >
> > [10] Zhou X.-X., Zeng W.-F., Chi H., Luo C., Liu C., Zhan J., He S.-M. & Zhang Z. (2017). pDeep: Predicting MS/MS Spectra of Peptides with Deep Learning. Anal. Chem., 89, 12690−12697
> >
> > [11] Zeng W.-F., Zhou X.-X., Willems S., Ammar C., Wahle M.,Bludau I., Voytik E., Strauss M. T. & Mann M. (2022). AlphaPeptDeep: a modular deep learning framework to predict peptide properties for proteomics. Nature communications, 13, 7238
> >
> > [12] Liu K., Tao C., Ye Y. & Tang H. (2024). SpecEncoder: deep metric learning for accurate peptide identification in proteomics. Bioinformatics, 40, i257–i265
> >
> > [13] Eloff, K.; Kalogeropoulos, K.; Morell, O.; Mabona, A.; Jespersen, J. B.; Williams, W.; Beljouw, S. P. B.; van Skwark, M.; Laustsen, A. H.; Brouns, S. J. J.; Ljungars, A.; Schoof, E. M.; Goey, J. V.; Keller, U. auf dem; Beguir, K.; Carranza, N. L.; Jenkins, T. P. (2023). De Novo Peptide Sequencing with InstaNovo: Accurate, Database-Free Peptide Identification for Large Scale Proteomics Experiments. bioRxiv,  DOI: 10.1101/2023.08.30.555055
> >
> > [14] Schmidt T., Samaras P., Dorfer V., Panse C., Kockmann T., Bichmann L., van Puyvelde B., Perez-Riverol Y.,  Deutsch E. W., Kuster B. & Wilhelm M. (2021). Universal Spectrum Explorer: A Standalone (Web-)Application for Cross-Resource Spectrum Comparison. J. Proteome Res., 20, 3388−3394

---

> > ### Comment · Reviewer_9C1C · 2024-08-24
> >
> > Thank you for addressing my feedback. I am satisfied with the described changes and have no further comment.

---

### Official Review · Reviewer_56xj · 2024-08-13
**A novel dataset focused on post-translational modifications (PTMs) in peptide sequences**

**Rating:** 7
**Confidence:** 4
**Correctness:** Yes.
**Clarity:** The paper is well-written and clear.

**Review:**

Quality: The paper presents a well-structured and comprehensive dataset that addresses a significant gap in the field of proteomics, specifically regarding post-translational modifications (PTMs) of peptides. The methodology for data collection, processing, and annotation is clearly described. The benchmarking results provided demonstrate the dataset's utility in improving machine learning model performance, indicating a high level of rigor in the research.

Clarity: The writing is clear, with a logical flow that guides the reader through the introduction, methodology, results, and conclusions. However, some sections could benefit from additional detail or examples to further clarify complex concepts.

Originality: The work is original in its approach to creating a rich, labeled dataset specifically focused on modified peptides, which is a relatively underexplored area in machine learning applications within proteomics.

Significance: The significance of this work is substantial, as it provides a valuable resource for researchers in proteomics and bioinformatics.

Pros:

Comprehensive Dataset: The dataset includes a large number of unique modified peptide sequences, enhancing its utility for various research applications.
Methodological Rigor: Clear description of data collection and annotation processes, allowing for reproducibility.
Benchmarking Results: Provides evidence of the dataset's effectiveness in improving machine learning model performance.
Addressing a Gap: Fills a critical gap in high-quality reference data for modified peptides, which is essential for advancing research in proteomics.

Cons:

Limited Scope of Modifications: While the dataset is extensive, it may not cover all possible PTMs, which could limit its applicability in certain contexts.
Over-Annotation Issues: The authors acknowledge potential over-annotation in the dataset, which could affect the accuracy of model training.
Complexity of Data Usage: Some users may find it challenging to integrate the dataset with existing tools and workflows without additional guidance.
Potential for Bias: The dataset is based on specific experimental conditions, which may introduce biases that could affect generalizability.

**Strengths:**

See above.

**Additional Feedback:**

None.

**Documentation:**

Yes.

**Ethics:**

No ethical concerns.

**Limitations:**

Yes.

**Opportunities For Improvement:**

See above.

**Relation To Prior Work:**

Yes, the submission clearly discusses how this work differs from previous contributions in several key ways.

**Summary And Contributions:**

The study highlights the challenges faced in the field due to the lack of high-quality reference data for modified peptides, which has hindered the advancement of predictive models. The authors provide a detailed description of the dataset's construction, including the types of modifications included and the methodologies used for data collection and annotation. They also present benchmarking results that demonstrate the utility of the dataset in improving model performance for tasks related to mass spectrometry.

---

> ### Author Rebuttal · Authors · 2024-08-16
>
> We thank the reviewer for highlighting areas where we can improve our work and providing valuable feedback. We want to address the comments by providing some clarifications and mentioning the changes we will implement in the final revision of the paper.
>
> ----
>
> **Limited Scope of Modifications**
>
> This is an important observation, and we would like to clarify further. First, we have data examples of the most common and most relevant PTMs, as seen in our Appendix Figure 3. While there are more than 400 PTMs [1], only some can be synthesized efficiently on a large scale. Using experimental data for others bears the risk of generating a training set with an unknown number of false positives. With the introduced datasets, researchers can explore ways to extract domain-specific features from PTMs (e.g. based on atom count) to train models and generalize on modifications not present in our dataset.
>
> We will clarify this further in the Limitations section in the revision of the paper.
>
> ----
>
> **Over-Annotation Issues**
>
> This is a valid limitation when predicting all the annotated fragment ions. Therefore, we trained and evaluated our models on a selected subset of the fragment ions (b and y ions without neutral losses). The models achieved a performance that is useful for downstream tasks, as reported in the Evaluation section, in published research [2,3,4,5], and as we show in the attached PDF in Figure 1 for an example downstream rescoring task.
>
> While predicting y/b ions is one possible formulation of the fragment intensity task, our datasets enable various other formulations, including predicting other ions or even the full experimental spectrum. Particularly, the latter would not suffer from over-annotation.
> We will clarify this further in the Limitations section in the revision of the paper.
>
> ----
>
> **Complexity of Data Usage**
>
> We provide the following approaches to enable accessibility to the datasets, as shown in Figure 1 in the Appendix (also in Figure 3 in the attached PDF with improvements for clarity):
> - a ready-to-use dataset collection hosted on the Hugging Face Hub [4] that includes a preprocessed and split dataset for each of the three tasks we discuss in the manuscript.
> - Zenodo datasets to have access to the full annotated data and an accompanying GitHub repository [8] with scripts to easily download, filter, and analyze the datasets.
>
> We opted for Hugging Face Hub since it is the machine learning community standard for sharing datasets and making them accessible. For the large annotated data, we chose Zenodo since it is an open repository that is commonly used to host large research datasets. Since Zenodo does not provide a programmatic way to download the data, we provided the GitHub repository [8] to facilitate downloading the data. With these resources available for users, we facilitate using the datasets for training, evaluating, and running pre-trained models (using tools such as DLOmix [11] and Koina [9]) and for downstream tasks (using a pipeline such as Oktoberfest [10]) - essentially the full pipeline.
>
> We will highlight this more clearly and possibly move the Figure to the main manuscript.
>
> ----
>
> **Potential for Bias**
>
> These are good points to highlight. We will add this to the limitations and highlight the bias. For the experimental setup (e.g. mass spectrometer used), we will elaborate on this in the text to mention the limitations of the data. Briefly, although the mass analyzers used for the acquisition of the datasets are only covering Orbitraps and Iontraps, our experiments showed that models trained on those (e.g. Prosit) generalize reasonably well to e.g. TOF and lead to similar increases in peptide identifications [5]. Also, since the physical peptides used to generate the reference data are still available, we expect further data from other peptide sets (e.g. PTMs) and mass spectrometers (e.g. Waters) to be added over the next years, further reducing biases and covering additional experimental settings. To show this, we have added a recently acquired timsTOF fragment ion intensity dataset to HuggingFace [6]. This dataset was acquired using the same physical peptides as the Orbitrap and Iontrap data.
>
> ----
>
> **More Details and Clarity**
>
> Thank you for the comment; we will iterate over the text once more and introduce clarifications where necessary in the final revision of the paper.
>
> Thanks again for your valuable review, and we appreciate any further elaboration or feedback.
>
> Note: references will be in a separate comment

---

> > ### Author Response · Authors · 2024-08-16
> > **References for Rebuttal**
> >
> > [1] Ramazi, S., & Zahiri, J. (2021). Post-translational modifications in proteins: resources, tools and prediction methods. Database, 2021, baab012.
> >
> > [2] Gessulat, S., Schmidt, T., Zolg, D. P., Samaras, P., Schnatbaum, K., Zerweck, J., ... & Wilhelm, M. (2019). Prosit: proteome-wide prediction of peptide tandem mass spectra by deep learning. Nature methods, 16(6), 509-518.
> >
> > [3] Gabriel, W., The, M., Zolg, D. P., Bayer, F. P., Shouman, O., Lautenbacher, L., ... & Wilhelm, M. (2022). Prosit-TMT: deep learning boosts identification of TMT-labeled peptides. Analytical Chemistry, 94(20), 7181-7190.
> >
> > [4] Wilhelm, M., Zolg, D. P., Graber, M., Gessulat, S., Schmidt, T., Schnatbaum, K., ... & Kuster, B. (2021). Deep learning boosts sensitivity of mass spectrometry-based immunopeptidomics. Nature communications, 12(1), 3346.
> >
> > [5] Adams, C., Gabriel, W., Laukens, K., Picciani, M., Wilhelm, M., Bittremieux, W., & Boonen, K. (2024). Fragment ion intensity prediction improves the identification rate of non-tryptic peptides in timsTOF. Nature communications, 15(1), 3956.
> >
> > [6] https://huggingface.co/datasets/Wilhelmlab/timstof-ms2
> >
> > [7] HuggingFace PROSPECT PTMs Dataset collection: ​​https://huggingface.co/collections/Wilhelmlab/prospect-ptms-665db48431a7e844634660ba
> >
> > [8] PROSPECT PTMs: https://github.com/wilhelm-lab/PROSPECT
> >
> > [9] Lautenbacher, L., Yang, K., Kockmann, T., Panse, C., Chambers, M., Kahl, E., ... & Wilhelm, M. (2024). Koina: Democratizing machine learning for proteomics research. bioRxiv, 2024-06.
> >
> > [10] Picciani, M., Gabriel, W., Giurcoiu, V. G., Shouman, O., Hamood, F., Lautenbacher, L., ... & Wilhelm, M. (2024). Oktoberfest: Open‐source spectral library generation and rescoring pipeline based on Prosit. Proteomics, 24(8), 2300112.
> >
> > [11] DLOmix: Python framework for Deep Learning in Proteomics https://github.com/wilhelm-lab/dlomix

---

### Decision · Program_Chairs · 2024-09-26

**Decision:**

Accept (Poster)

**Comment:**

The paper presents a comprehensive dataset aimed at enhancing machine learning applications in proteomics, specifically for predicting properties of post-translationally modified peptides. The dataset, named PROSPECT PTMs, includes 1.2 million unique modified peptide sequences and 30 distinct amino acid-PTM pairs. This work addresses a significant gap in high-quality reference data for modified peptides, which has previously impeded the development of predictive models in mass spectrometry-based proteomics.

All reviewers agree that this work addresses a crucial gap in proteomics research by providing a comprehensive, high-quality dataset for modified peptides. This has the potential to drive advancements in various downstream tasks, from machine learning model training to peptide identification and spectral library generation. The paper is well-organized and clearly written, making complex topics more accessible. It provides thorough documentation and benchmarks, which highlight the utility and broad applicability of the dataset.

The authors employed a meticulous annotation and validation process for the dataset, ensuring that it is reliable and suitable for training high-performance machine learning models. This careful design will help ensure the dataset’s wide adoption in proteomics research. Despite using synthetic data, the dataset is derived from experimentally observed peptides, which enhances its generalizability. This ensures that models trained on this dataset can apply effectively to real-world biological samples, a point that the authors have addressed convincingly in their rebuttals. By hosting the dataset on widely used platforms like Hugging Face and Zenodo, and providing accompanying tools for easy access and use, the authors have ensured that this dataset is not only useful but also accessible to the broader research community.

While the dataset has been rigorously tested on synthetic data, demonstrating its performance on more natural biological datasets could further solidify its impact. This could help alleviate concerns about potential biases introduced by synthetic peptide data. While the authors provide scripts for easy access to the datasets, offering additional user-friendly tools and clear examples for broader use cases could help researchers from different backgrounds maximize the dataset's potential.

Overall, this paper makes a important contribution to the field of proteomics, offering a novel dataset that will likely become a key resource for advancing machine learning applications involving PTMs. The work is well-executed and clearly presented, and it addresses a gap that has been widely recognized by the community. The few limitations that exist can be addressed in future work, but they do not detract from the significance of this contribution.